



# Accurate loads and velocities on low solidity wind turbines using an improved Blade Element Momentum model

Yassine Ouakki[1] and Abdelaziz Arbaoui[1]

[1]INSCM team, LM2I, National School of Arts and Crafts (ENSAM), Moulay Ismail University, BP 4024 Meknes, Morocco

**Correspondence:** Yassine Ouakki (yassineouakki@gmail.com)

**Abstract.**

The accurate prediction of loadings and velocities on a wind turbine blades is essential for the design and optimization of wind turbines rotors. However, the classical BEM still suffer from an inaccurate prediction of induced velocities and loadings, even if the classical correction like stall delay effect and tip loss correction are used. For low solidity rotors, the loadings are generally over-predicted in the tip region, since the far wake expansion is not accurately accounted for in the one-dimensional (1D) momentum theory. The 1D dimensional momentum theory supposes that the far wake axial induction is equal to twice the axial induction in the rotor plane, which results in an under-estimation of the axial induction factor in the tip region. Considering the complex nature of the flow around a rotating blade, the accurate estimation of 3D effects is still challenging, since most stall delay models still often tend to under-predict or over-predict the loadings near the root region. As for the solution method for the classical BEM equation, the induced velocities are computed accounting for the drag force. However, according to the Kutta-Joukowski theorem, the induced velocities on a blade element are only created by lift force. Accounting for drag force when solving the BEM will result in an over-estimation of the axial induction factor, while the tangential induction factor is under-estimated. To improve the accuracy of the BEM method, in this paper, the 1D momentum theory is corrected using a new far wake expansion model to take into account the radial flow effect. The blade element theory is corrected for three-dimensional effects through an improved stall delay model. An improved solution method for the BEM equations respecting the Kutta-Joukowski theorem is proposed. The improved BEM model is used to estimate the aerodynamic loads and velocities on the National Renewable Energy Laboratory Phase VI rotor blades. The results of this study show that the proposed BEM model gives an accurate prediction of the loads and velocities compared to the classical BEM model.

## 1 Introduction

Thanks to its simplicity and robustness, the blade element momentum (BEM) theory has been widely used in the wind industry for the design and optimization of wind turbine rotors (Tangler, 2002). It was first introduced by Glauert (1935), as a combination of the 1D momentum theory (MT) and the blade element theory (BET). The 1D MT, developed by Glauert (1935), is describing the process of energy extraction by the wind turbine using the conservation laws of fluid mechanics. The BET, developed by Froude (1878), is describing the local loadings and velocities computation on a blade using airfoil theory. Some assumptions in the BEM theory are corrected to make the prediction more realistic. The assumption of an infinite number of



blades in the 1D MT is corrected using the Prandtl (1921) tip and hub loss model. The three-dimensional effects (3D stall delay) are also taken into account by correcting the 2D aerodynamic coefficients in the BET. However, The BEM model still suffers from its inaccurate prediction of induced velocities and the non-dimensional loading.

The 1D MT equations are originally developed by Glauert (1935) by simplifying the general momentum theory (GMT)
. The first simplification was neglecting all nonlinear tangential velocity terms; as a result, the pressure drop due to wake rotation was neglected. The second simplification was using the GMT equation in differential form, which has been proven to be wrong by Goorjian (1972). Consequently, the axial induction in the far wake is taken as twice the induction at the rotor disc. Joukowski (1912) and Sharpe (2004) considered the effect of the pressure drop due to wake rotation, they found out that the rotor power coefficient increases at low tip speed ratio and can exceed the Betz-Joukowsky limit. However, The influence of
the pressure drop due to wake rotation is important for slow running rotors when the tip-speed-ratio is small (Vaz et al., 2011) . In contrast, For modern wind turbine rotors operating with hight rotational speed and low torque (low solidity, typically 7% or less), the rotational kinetic energy in the wake will be small (Gupta and Leishman , 2005). As a result, the pressure drop due to wake rotation can be neglected. But, the use of an erroneous differential form of the GMT, will result in the cancelation of wake expansion by the pressure drop due to wake rotation. The power loss from wake rotation at a low tip speed ratio will be
almost canceled by the increased mass flow through the rotor (De Vries (1979), Sharpe (2004), and Xiros and Xiros (2007)). However, at the level of spanwise loading, the wake expansion and pressure drop due to wake rotation do not cancel each other. In fact, The spanwise loading will be reduced at the tip region due to the presence of radial flow, and will be augmented at the hub region due to the pressure drop due to wake rotation ( Mast et al. (2004), Dossing et al. (2012), and Madsen et al. (2010)). Recently, Sun et al. (2016) proposed an improved formulation of GMT accounting for both effects and found out that
the axial induction factor in the far wake is always smaller than twice the axial induction at the rotor plane.

The classical BEM theory works well at low and moderate tip speed ratios, but it is not reliable at high tip speed ratios where the expansion of the wake is large (Sørensen et al. , 1998). Whale et al. (2000) through an experimental investigation found out that the wake expansion immediately behind the rotor becomes more pronounced towards high tip speed ratios. Carrión et al. (2015) performed a CFD analysis of the wake behind the Mexico rotor, They concluded that the higher the tip speed ratio, the
more expansion of the wake. Recently, Micallef et al. (2013) performed a numerical and experimental investigation of the radial flow close to the rotor plane. They showed that the radial velocity reaches important magnitudes especially in the tip region and it is predominantly outboard, as a result, the wake expands. As for the root region, they found that there is almost no expansion of the wake. Herráez et al. (2014) has shown that the high axial induction will lead to an important wake expansion caused by a significant radial flow, especially in the tip region. Madsen et al. (2010) performed a comparison between the classical
BEM and actuator disc. They found a close correlation between the AD and the classical BEM model for the integral value of the power coefficient. However, locally along the blade radius, they found that the classical BEM model overestimates the power coefficient and under-estimate the axial induction on the outboard part due to the wake expansion. Similarly, Johansen et al. (2004) found that a slightly lower induction in the tip region compared with the EllipSys3D computations. To correct the 1D MT, Madsen et al. (2010) proposed an empirical model to correct directly the induction for the expansion effect by
fitting the AD simulations. Recently, Sun et al. (2016) proposed a generalized formulation of MT accounting for radial flow.





However, in order to close the equation system, They used the semi-analytical AD equation of radial flow taken from the AD simulation by Madsen et al. (2010). Nevertheless, these BEM models are strongly coupled and require multiple iterative loops to be solved. As a result, they are computationally expensive compared to the classical BEM model, especially when used for the optimization of wind turbines rotors.

Considering the complexity of three-dimensional flow around a rotating blade, the non-dimensional 2D lift and drag, extracted from a wind tunnel for a given airfoil, fail to predict the aerodynamic loadings on a rotating blade at the inboard sections of the blades. In the last decades, extensive studies of this phenomenon were performed and several correction models were proposed (Spera (2009), Butterfield et al (1992), Lee and Wu (2013), Snel et al. (1993), and Breton et al. (2008)). These studies agree on the fact that for low wind speeds and angles of attack bellow the static stall angle, the flow remains attached to

the blade and there is a minor difference between the lift and drag of a rotating blade and a non-rotating blade. however, when the 3D stall occurs at hight angles of attack, the flow began to separate from the blade surface and rotate with the blade. As a result, flow is subjected to rotational forces namely centrifugal and Coriolis forces. The centrifugal force induces an outboard radial flow, thus, the radial flow induces Coriolis forces that act as a favorable pressure gradient in the chord-wise direction. These forces lead to a delay in the separation process, resulting in an increase in the maximum lift coefficient at a higher

angle of attack. This phenomenon is known as stall delay or rotational augmentation. However, based on the comparative study performed by Breton et al. (2008) on six existing stall delay models, they concluded that these models fail to predict the 3D behavior compared to NREL's phase VI experiment mainly because of their lack of generality since this phenomenon is not yet fully understood. However, compared to other stall delay models, Bak et al. (2006) model was found to have superior accuracy thanks to the modeling approach that was based on estimating the pressure gradient instead of estimating the gradient of the

force due to rotational effects. However, Bak et al. (2006) model is still over-estimating the loads. This original approach was adopted by Wang et al. (2013) to propose a new stall delay model by solving simplified Navier-Stocks equations for the pressure. Nevertheless, these models that are based on estimating pressure gradient are strongly empirical and showing a relatively good agreement with experimental data. However, Bak et al. (2006) model is not yet extensively validated, particularly, the force distributions along the blades (Breton et al. , 2008).

Moreover, The classical BEM model computes the induced velocities by solving iteratively the BEM equations accounting for the drag force. However, the induced velocities are produced by vortices (lift or circulation) according to vortex theory and particularly the Kutta-Joukowski theorem (De Vries , 1979). Thus, the induced velocities are only created by lift force. Lindenburg (2003) has shown that in the case of a non-swept rotor blade, the relative induced velocity on the airfoil is a result of the circulation around it. Thus, it is perpendicular to the local relative flow. Wilson and Lissaman (1974) pointed out that

the drag should be excluded from BEM equations because the drag-based velocity deficit is only a characteristic of the wake and it does not contribute to the induced velocity at the rotor disc. Recently, Shen et al. (2005) made a detailed analysis of BEM equations in the tip region. It was found that if we account for the drag force in the BEM equation, the relative velocity becomes zero at the tip, which is not physically true since the tip vortex is created at the blade tip and then converted into the wake. Given these arguments, it is still arguably if the drag should be excluded from the BEM equations when computing the





induced velocities ( Burton et al. (2001) and Hansen (2015) ). In either case, the drag force should be accounted for once the
      induced velocities are computed; since the drag force has an impact on the performance of a wind turbine rotor.

      This manuscript is organized as follows. In Section 2, the GMT is applied to low solidity wind turbine rotors, then a new
      far wake expansion model is proposed using dimensional analysis. In Section 3, the blade element model is described, then an
      improved stall delay model is proposed. In Section 4, An improved BEM model respecting the Kutta-Joukowski theorem is
given, the improved BEM equations are solved using the guaranteed convergence algorithm. In Section 5, A detailed validation
      of the proposed BEM model is performed based on the experimental results of the NREL Phase VI rotor (Hand et al , 2001).
      At last, the conclusion and further improvements of the BEM model will be given.

## 2    Improved momentum theory

### 2.1    General momentum theory applied to low solidity wind turbines

The general momentum theory (GMT) is based on a global description of the flow around the wind turbine rotor. The rotor is
      modeled as an actuator disc with an infinite number of blades. The flow is considered steady, incompressible and axisymmetric.
      The GMT take into account the radial, axial and azimuthal velocities. The axial and radial velocity are continuous, while the
      azimuthal velocity has a discontinuity at the actuator disc. A rotation of the flow is present in the wake, and it is absent upstream
      of the rotor disc. The pressure drop at the rotor disc is due to the change in azimuthal velocity at the rotor disc (Sørensen ,
2016). Assuming that the air masse contained in the rotor disc is separate from the rest of the air in the rotor plane, as a result,
      the air masse at the rotor disc can be contained in a circular boundary surface. It can be extended upstream and downstream
      of the rotor forming a long stream-tube of circular cross-sections. Since the air within the circular boundary of the stream-tube
      is assumed to be incompressible; the cross-sectional area of the stream-tube must expand to accommodate the drop in wind
      speed. Thus, the conservation laws of fluid mechanics can be applied globally along the stream tube or locally along an annular
stream tube (see Fig. 1).

      The annular element of the actuator disc is taken at a radial position $r$ and the actuator disc rotating with a rotational speed
      noted $\Omega$. The annular element at the far wake is taken at a radial position $r_w$ and rotating with a rotational speed noted $\omega_w$. Due
      to the discontinuity in the rotational motion of the flow at the rotor plane, The tangential velocity at the rotor disk is taken as
      the average of the upstream and downstream plane $V_t = -r\Omega a'$. Since there is no rotation of the flow in the upstream plan, the
tangential velocity is zero upstream of the rotor $V_{t+} = 0$ , the tangential velocity straight after the rotor $V_{t-} = -2r\Omega a'$ , and
      $V_{tw} = -r_w\omega_w$ in the far-wake. In contrast, the axial and radial velocities are continuous, thus the radial and axial velocities
      straight after and before the rotor are identical. The radial velocity in the far upstream and downstream plans is zero since
      there is no expansion of the far upstream and downstream wake. In the rotor plane, the radial velocity is noted $U_{rad}$. The axial





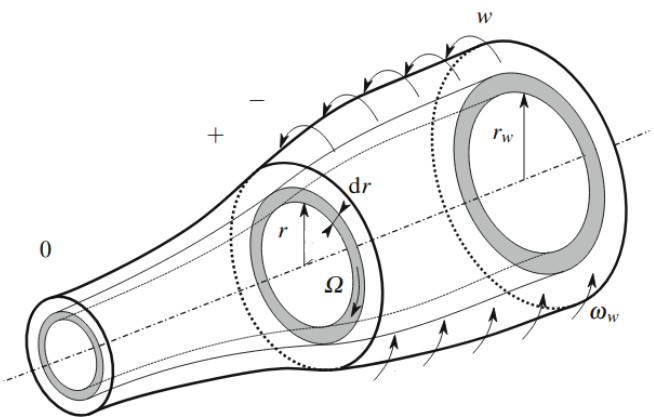

**Figure 1.** Actuator disc concept

velocity in the far upstream is equal to the free stream $U_0$. The axial velocity in the rotor plan $U_n$ and The axial velocity in the

far wake $U_{nw}$ are defined by Eq. (1) and Eq. (2), respectively.

$$U_n = U_0(1 - a) \tag{1}$$

$$U_{nw} = U_0(1 - b) \tag{2}$$

Where $a$, $a'$, and $b$ are: the axial induction at the rotor plan, tangential induction at the rotor plane, and axial induction at the far wake, respectively.

Applying the conservation laws under the assumption of the general momentum theory, the governing equations can be found. The thrust (Eq. (3)) and torque (Eq. (4)) at a given radius of the rotor disc are given by (Sørensen , 2016). The GMT accounts for the pressure drop due to wake rotation, the wake expansion, and the pressure contribution to the lateral force component.

$$C_T = 2b(1 - a) + \frac{1 - a}{1 - b}\frac{p_0 - p_w}{1/2\rho U_0^2} + \frac{1}{1/2\rho U_0^2}\frac{dT_{p,Side}}{dA} \tag{3}$$

$$C_Q = 4a'(1 - a)\lambda_r \tag{4}$$



Where, $dT_{p,Side}$ is the pressure contribution to the lateral force component, $dA$ is the elementary disk area, and $\rho$ is the air density. $p_w$ and $p_0$ are the pressure in the far wake and the atmospheric pressure respectively. $\lambda_r = r\Omega/U_0$ is the speed ratio.

The thrust coefficient in the GMT does not form a closed and solvable system because the unknowns number is too high compared to the number of the equations. Consequently, additional assumptions need to be introduced. The lateral force component, $dT_{p,Side}$ is difficult to determine and can be found using CFD, however, this term is generally considered to be small (Sørensen , 2016). Thus, the annular element are independent of each other. Additionally, modern horizontal axis wind turbines (HAWT) are operating at high rotational speed in order to minimize the wake rotation losses and thus extract more power (Gupta and Leishman , 2005). As a result of high rotational speed, the rotor will have less solidity than a rotor operating at low rotational speed (Burton et al., 2011). Additionally, low solidity wind turbines ( typically 7% or less) will result in a low manufacturing cost of the blades (Tangler (2000) and Burton et al. (2011)). As a result, the pressure drop due to wake rotation can also be neglected ($p_0 = p_w$). The thrust coefficient becomes similar to 1D MT except for the expansion effect that is parameterized by the far wake expansion ratio ($\chi = b/a$) to be determined instead of taking $b = 2a$ as the 1D MT. The hypothesis of an infinite number of blades is corrected using the Prandtl tip and hub loss model (Prandtl, 1921) . The thrust and torque are given by Eq. (5) and Eq. (6), respectively.

$$C_T = 2\chi a F(1-a) \tag{5}$$

$$C_Q = 4a'F(1-a)\lambda_r \tag{6}$$

Prandtl tip and hub loss model $F = F_{tip}F_{hub}$ is given by Eq. (7) and Eq. (8), respectively.

$$F_{tip} = \frac{2}{\pi}cos^{-1}(exp[-\frac{N}{2}\frac{R-r}{rsin(\phi)}]) \tag{7}$$

$$F_{hub} = \frac{2}{\pi}cos^{-1}(exp[-\frac{N}{2}\frac{r-r_h}{r_h sin(\phi)}]) \tag{8}$$

Where $\phi$ is the inflow angle, $R$ is the tip radius, $r_h$ is the hub radius, and $N$ is the number of blades.

Note that the radial flow is absent in the thrust coefficient equation because of the absence of radial flow in the far upstream and downstream to the actuator disc and it is only present in the near wake, reaching a maximum at the rotor disc when following a streamline (Van Kuik , 2017a). This was proved by Van Kuik (2017a) using vorticity-based arguments. As a result, When applying the conservation of momentum, the radial flow is canceled out, but its effect remains present in the far wake by the far wake expansion ratio.





## 2.2 New far Wake expansion model

The far wake expansion radius can be found by applying the conservation of the axial flow between the rotor disc and the far wake. It is given by Eq. (9).

$$\frac{r_w}{r} = \sqrt{\frac{1-a}{1-b}} \tag{9}$$

In case of the 1D MT, the far wake axial induction is taken as twice the axial velocity in the rotor plan (b=2a), the 1D far wake expansion radius is given by Eq. (10).

$$\frac{r_w}{r} = \sqrt{\frac{1-a}{1-2a}} \tag{10}$$

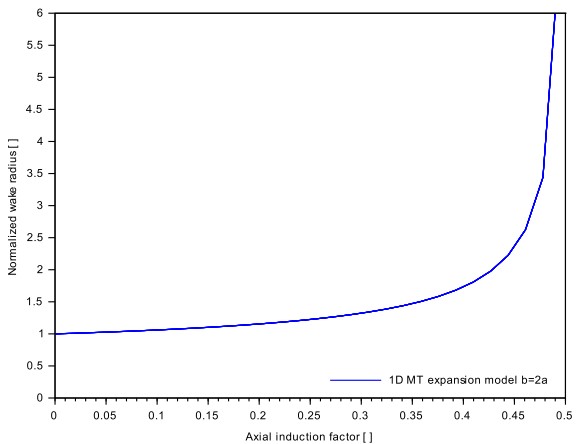

**Figure 2.** 1D MT Far wake expansion radius a a function of axial induction factor

One of the major drawbacks of the 1D MT far wake expansion model is the unrealistic expansion when the axial induction is approaching 0.5 as the wake velocity tends to zero as shown in Fig. 2. For this reason, the MT is replaced by an empirical
equation for axial induction higher than a critical value between 0.2 and 0.4 depending on the empirical model used (Pratumnopharat et al, 2011). Low solidity wind turbines are characterized by small wake expansion compared to high solidity rotors (Sezer-Uzol and Uzol (2013), Sørensen and Kock (1995), and Zahle and Sørensen (2007)).Thus for a given axial induction factor, the far wake expansion radius for low solidity rotors will be further over-estimated. Consequently, the 1D MT will be valid only for a very low axial induction factor which corresponding to a low tip speed ratio. This is generally because
the far wake expansion ratio in 1D MT is independent of operating conditions and geometry of the wind turbine rotor.





To take into account the operating conditions and rotor's geometry effects on the far wake expansion ratio. First, the fractional decrease in axial wind speed between the rotor plane and the far wake is defined by Eq. (11). It is a normalized and dimensionless quantity between 0 and 1. In case of no expansion of the wake $\xi = 0$, and the limiting case $\xi = 1$ corresponds to the maximum wake expansion ratio ( b=2a) since $b/a < 2$ according to GMT. Second, the classical dimensional analysis
(Graebel , 2001) is used to identify all relevant dimensional grouping to the far wake expansion, then applying the Buckingham pi theorem to estimate the fractional decrease in axial wind speed $\xi$. Hence, the far wake expansion ratio can be computed by $\chi = \xi - 1$.

$$\xi = \frac{b-a}{a} \tag{11}$$

All dimensional parameters that are relevant to the far wake expansion are identified. They are shown in Table 1. The actuator
disc geometry parameters include the blade tip radius $R$, the local radial station $r$, and the blade number $N$. The flow conditions are the free stream speed $U_0$, the rotor rotational speed $\Omega$, and the additional velocity induced by the wake $\nu$ (m/s). The classical Buckingham pi theorem states that the number of dimensionless groupings is equal to the number of dimensional parameters subtracted by the number of independent dimensions (length, time). Note that the blade number is already dimensionless. thus, there are a total of three dimensionless groupings: the normalized blade radius, the tip speed ratio, and additional velocity
induced by the wake normalized by the free stream. The blade number is the fourth dimensionless number (Table 2).

**Table 1.** Parameters relevant to far wake expansion ratio.

| Blade geometry | Flow conditions |
|---|---|
| r   R   N | $U_0$   $\Omega$   $\nu$ |

**Table 2.** Dimensionless groupings relevant to far wake expansion ratio.

| Blade geometry | | Flow conditions | |
|---|---|---|---|
| $\frac{r}{R}$ | N | $\lambda = \frac{R\Omega}{U_0}$ | $\bar{\nu} = \frac{\nu}{U_0}$ |

The above dimensionless grouping can be further reduced to one dimensionless group that characterizes the wake, which is the apparent helical pitch (Eq.12) relating the rotor parameters $(N, \lambda, r/R)$ to the far wake parameters $(\bar{h}, \bar{\nu})$.

$$\bar{h} = \frac{h}{Nr} = \frac{2\pi}{N(r/R)\lambda}(1 - \bar{\nu}) \tag{12}$$

Where, $h$ is the helical pitch of the wake, and $\bar{h}$ is the normalized apparent helical pitch of the wake.



This finding is in accordance with vortex theory, where the far wake parameters $(r_w, h, \nu)$ are related to the near wake $(r, R, N, \lambda, C_T)$ by an expansion model (Okulov et al. (2015)). The thrust coefficient is excluded in this work since it is already depending on the wake expansion and also to avoid a strong coupling of BEM equations. Thus, the dimensionless expansion radius $r_w/r$ depends on the apparent helical pitch $\bar{h}$, dimensionless velocity induced by the wake $\bar{\nu}$. However, according to Okulov et al. (2015), for small expansion of the wake, the additional velocity induced by the wake of the tip vortices equals

half the averaged induced axial velocity with a correction of a small expansion ($\epsilon = r_w - r$) as shown in Eq.13. As a result, the velocity induced by the wake has a small variation ($0 \leq \bar{\nu} \leq 0.25(1 + \epsilon)$), when the rotor is operating in the momentum region ($0 \leq a \leq 0.5$), compared to the apparent helical pitch varying from zero to infinity. Thus, the additional velocity induced by the wake can be neglected and the apparent helical pitch is considered as the most important dimensional parameter.

$$\bar{\nu} = \frac{a}{2}(1 + \frac{\epsilon}{R}) \tag{13}$$

Before applying the Buckingham pi theorem to estimate the fractional decrease in axial wind speed, the normalized apparent helical pitch should be normalized by the relative distance $\sqrt{r^2 + h^2}$ instead of the blade radius only. According to the pi theorem, the fractional decrease in axial wind speed is given by Eq. (14). As a result, the far wake expansion ratio is given by Eq. (15).

$$\xi = a_0(\frac{h/N}{r} \frac{r}{\sqrt{(r^2 + (h/N)^2)}})^{b_0} \tag{14}$$

$$\chi(r) = 1 + a_0(\frac{\bar{h}}{\sqrt{1 + \bar{\bar{h}}}})^{b_0} \tag{15}$$

Where $a_0$ and $b_0$ are empirical parameters. In this paper, these parameters are taken as $a_0 = 1$ and $b_0 = 1$ for the NREL Phase VI rotor. Once the far wake expansion ratio is found, the new far wake expansion radius can be computed using Eq.16 as follows:

$$\frac{r_w}{r}(r) = \sqrt{\frac{1 - a(r)}{1 - a(r)\chi(r)}} \tag{16}$$

## 3    Blade element model accounting for stall delay effect

### 3.1    Blade element theory

The blade element theory assumes that the blades are made up of a number of blade elements arranged in the spanwise direction, so the airfoil theory can be used to compute the local loads and velocities acting on each blade element. The BET assumes





that the blade elements are aerodynamically independent and do not have any interference between them. The loads can be
obtained from the 2D lift, drag and moment coefficients of the airfoil at any radial position. The inflow is known at the blade
and a lifting-line assumption is used (Branlard , 2017). As a result of these assumptions, the global loads on a blade can be
integrated over the total blade span incorporating the velocity terms, to obtain the thrust, the torque, and power developed by
the blade. This is further multiplied by the number of blades to get the total rotor thrust, torque, and power.

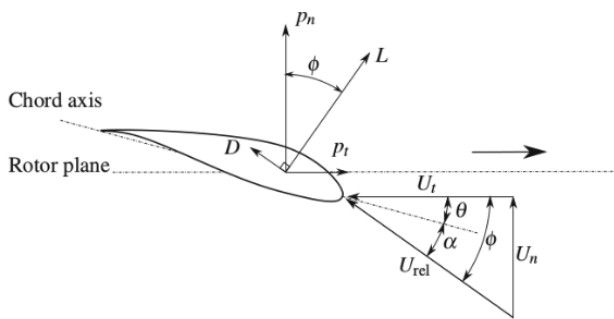

**Figure 3.** Velocity triangle and resulting aerodynamic foeces applied to an airfoil.

A blade element is an infinitesimal fraction dr of the blade radius R at a radial position r, so it can be considered as an
airfoil. An airfoil is defined with by its chord c(r), its twist $\theta(r)$ and its shape. The relative wind applied to a blade element,
noted $U_{rel}(r)$, is decomposed into a normal component $U_n(r)$ and a tangential component $U_t(r)$ to the rotor plane. Three
characteristic angles defined by the velocity axis and the chord axis: the flow angle $\phi(r)$ between the tangential and relative
velocity and it is assumed to be known, the airfoil twist $\theta(r)$ about to the rotor plane, and the angle of attack $\alpha(r)$ between the
relative velocity and the chord axis and it is related to aerodynamic loads. Since the lifting-line assumption is used, the velocity
triangle can be defined in terms of the axial and tangential inductions factors $a$ and $a'$ (Eq. 17 and Eq.18). The relationship
between these angles and the velocity components are given in Equations 19 and 20. The blade element, velocity triangle, and
aerodynamic forces are shown in Figure 3.

$$U_n = U_0(1 - a) \tag{17}$$

$$U_t = \Omega r(1 + a') \tag{18}$$

$$tan(\phi) = \frac{U_n}{U_t} \tag{19}$$



$$\alpha = \phi - \theta \tag{20}$$

The lift and drag forces per unit of length applied on the blade element of the surface are defined as follows:

$$L = 1/2\rho c U_{rel}^2 C_{l2D} \tag{21}$$

$$D = 1/2\rho c U_{rel}^2 C_{d2D} \tag{22}$$

Where $(C_{l2D}(\alpha), C_{d2D}(\alpha))$ are the 2D lift and drag coefficients respectively. The normal and tangential forces per unit of length are defined as the projection of the lift and drag on the longitudinal plane and rotor plane (Fig. 3). They can also be defined in the function of normal and tangential coefficients.

$$P_n = 1/2\rho c U_{rel}(C_{l2D}cos(\phi) + C_{d2D}sin(\phi)) = 1/2\rho c U_{rel}^2 C_{n2D} \tag{23}$$

$$P_t = 1/2\rho c U_{rel}^2(C_{l2D}sin(\phi) - C_{d2D}cos(\phi)) = 1/2\rho c U_{rel}^2 C_{t2D} \tag{24}$$

Since the wind turbine rotor has identical blades, the loading on the blade elements of the blades at the same radius will be also identical. The local thrust, torque and power coefficients applied to an elementary annulus at a radial position r of the rotor disc are defined as follows:

$$C_T = \frac{NP_n dr}{1/2\rho U_0^2 2\pi r dr} = \frac{(1-a)^2}{sin^2(\phi)}\sigma C_{n2D} \tag{25}$$

$$C_Q = \frac{NP_t r dr}{1/2\rho U_0^2 r 2\pi r dr} = \frac{(1-a)(1+a')}{sin(\phi)cos(\phi)}\lambda_r \sigma C_{t2D} \tag{26}$$

$$C_P = C_Q\lambda_r = \frac{(1-a)(1+a')}{sin(\phi)cos(\phi)}\lambda_r^2 \sigma C_{t2D} \tag{27}$$

where $\sigma = Nc/(2\pi r)$ is the local solidity.



### 3.2 3D stall delay modeling

The stall delay phenomenon is still challenging current 1D models as it was shown by Breton et al. (2008) by performing a comparative study of several stall delay model. However, Bak et al. (2006) model was found to have superior accuracy thanks to the modeling approach that was based on estimating the pressure gradient due to rotation instead of estimating the loading gradient. Although, Bak model is still over-estimating the loads. In addition, it is not yet fylly validated for the spanwise distribution of the loadings. In this work, The stall delay model originally developed by Bak et al. (2006) will be improved based on recent advances in this subject.

#### 3.2.1 Bak model

Bak et al. (2006) was the first to develop a stall delay model that is based on estimating the pressure gradient instead of the force gradient due to the blade rotation. The pressure gradient is defined as the product of two functions: An amplification and a shape. The amplification was estimated as the ratio of the sum of centrifugal and Coriolis force to the pressure force, these forces were estimated through an order of magnitude analysis of NS equations. The shape was estimated through a simple empirical equation based on fitting the 3D experimental data of NREL phase VI at 30% of the blade radius. The pressure gradient is given by Eq. 28.

$$\Delta C_P = 5(1 - \frac{x}{c})(\frac{\alpha - \alpha_0}{\alpha_1 - \alpha_0})^2 \sqrt{1 + (\frac{R}{r})^2} \frac{c}{r} \frac{1}{1 + tan^2(\phi)} \tag{28}$$

Where $\alpha_0$ is the AoA at which the flow begins to separate and $\alpha_1$ is the AoA at which the flow is completely separated. For the S809 airfoil used in the NREL phase VI rotor. $\alpha_1$ and $\alpha_2$ can be approximately taken as $6.2^o$ and $21^o$. x/c is the normalized chord-wise position.

The normal and tangential forces gradient is found by integrating the pressure gradient over the full chord.

$$\Delta C_n = \int\limits_{x/c=0}^{x/c=1} \Delta C_P d(\frac{x}{c}) \tag{29}$$

$$\Delta C_t = \int\limits_{y/c=y/c(leading-edge)}^{y/c=y/c(trailing-edge)} \Delta C_P d(\frac{y}{c}) \tag{30}$$

The 3D normal and tangential force coefficients are computed by

$$C_{n3D}(\alpha) = C_{n2D}(\alpha) + \Delta C_n \tag{31}$$

$$C_{t3D}(\alpha) = C_{t2D}(\alpha) + \Delta C_t \tag{32}$$





### 3.2.2 Improved Bak model

The stall delay model proposed by Bak assume that the chordwise velocity is zero and the spanwise velocity is dominant

when the flow is separated. This hypothesis was also adopted by Corten (2001) based on an experimental study on the flow separation on the wind turbine blade. As a result, this model is only valid inside the separated area where the radial flow is constant. Assuming that the pressure gradient induced by the spanwise flow in the separation area mainly affects the normal force of the local airfoil, since the pressure gradient depends only on the chord-wise position. This is similar to the analysis of (Lindenburg , 2003). The normal force gradient can be computed by integrating the pressure gradient in the separated area is

given by Eq. 33.

$$\Delta C_n = \int_{x/c}^{1} \Delta C_P d(\frac{x}{c}) = \frac{5}{3} f^3 (\frac{\alpha - \alpha_0}{\alpha_1 - \alpha_0})^2 \sqrt{1 + (\frac{R}{r})^2} \frac{c}{r} \frac{1}{1 + tan^2(\phi)} \tag{33}$$

Where $f = 1 - x/c$ is the trailing edge separation factor. It can be estimated using the Kirchhoff-Helmholtz model (Eq. 34) that is commonly used in the dynamic stall computation (Leishman and Beddoes , 1986).

$$f = 1 - \left( 2 \sqrt{\frac{C_{n2D}}{(\alpha - \alpha_0) \frac{\partial C_{n2D}}{\partial \alpha}}} - 1 \right)^2 \tag{34}$$

Note that the improved bak model is in good agreement with Wang et al. (2013) findings. Since, Wang et al. (2013) solved the inviscid stall delay model developed by Corten (2001). They found that the shape of the normal force gradient to be a third-order plynomial as a function of the trailing edge separation factor, which is consistent with Eq. 33.

### 3.2.3 Performance tip loss correction

The blade element theory does not take into account the finite span of the blade; it treats all sections of the blade the same way.

In reality, this is not true because there is always aerodynamic loss at the tip of the blade (Shen et al. , 2005). The performance tip loss is used to correct the 3D aerodynamic force coefficients; since the forces should converge to zero at the tip to equalize the pressure between the upper and lower surface of the blade. Shen's tip loss model (Shen et al. , 2005) is used in this paper. It is given by Eq. (35).

$$F_{1tip} = \frac{2}{\pi} cos^{-1}(exp[-h(r) \frac{N}{2} \frac{R - r}{r sin(\phi)}]) \tag{35}$$

$$h_0 = exp[-c_1(N\lambda - c_2)] + 0.1 \tag{36}$$

The correction function $h_0$ is depending only on the number of blades and the tip speed ratio. The constants $c_1$ and $c_2$ are determined from experimental data. In case of the NREL phase VI rotor: $c_1 = 0.125$ and $c_2 = 21$.





The 2D aerodynamic data used in this work are taken from Lindenburg (2003) for the non-rotating blade. The performance tip loss model will be used in both Bak and improved Bak models. The 3D correction of the lift and drag are given by Eq. (37) and Eq. (38).

$$C_{l3D}(\alpha) = F_{1tip}(C_{l2D}(\alpha) + \Delta C_n cos(\alpha)) \tag{37}$$

$$C_{d3D}(\alpha) = F_{1tip}(C_{d2D}(\alpha) + \Delta C_n sin(\alpha)) \tag{38}$$

## 4 Improved BEM model

In order to compute the BEM solution $(a, a', \phi)$, the improved BEM model needs to be completed by respecting the KJ theorem and using a wake state model for axial induction factor $a > a_c$. Then, the improved BEM equations will be solved using the guaranteed convergence algorithm to avoid any convergence issues.

### 4.1 Induced velocities computational method

The conceptual model of Kutta-Joukowski for the wind turbine rotor is identical to the BET, except for the exclusion of drag force. Since, the induced velocities are produced only by lift force according to vortex theory and particularly the Kutta-Joukowski theorem (Okulov et al. , 2015). As a result, The orthogonality condition must be respected between the total lift force and the relative velocity (see Fig. 4).

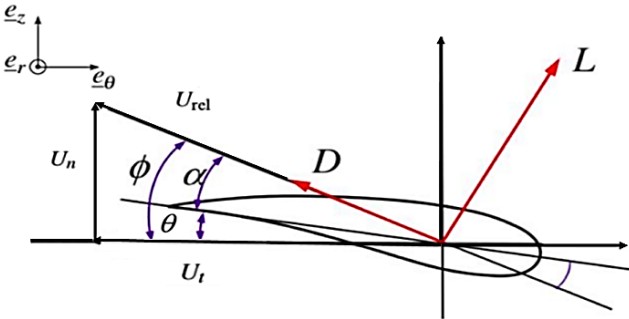

**Figure 4.** KJ Velocity triangle and resulting aerodynamic foeces applied to an airfoil.

By applying The Kutta-Joukowski theorem to a blade element of length dr, the total force is given by Eq. (39).

$$\boldsymbol{L_{KJ}} = \rho U_{rel} \boldsymbol{\Gamma} \tag{39}$$





Where, $\Gamma$ is the circulation vector defined by Eq. (40).

$$\mathbf{\Gamma} = 0.5 c U_{rel} \mathbf{C_L} \tag{40}$$

The local thrust and torque produced at an elementary annulus at a radial position r of the rotor are obtained from the superposition of the local contributions $L_{KJ}$ of each of N blades. The local thrust and torque coefficients applied to an elementary annulus are given by Eq. (41) and Eq. (42).

$$CT_{KJ} = \sigma \frac{(1-a)^2}{sin^2(\phi)} C_{l3D} cos(\phi) \tag{41}$$

$$CQ_{KJ} = \frac{(1-a)(1+a')}{sin(\phi)cos(\phi)} \lambda_r \sigma C_{l3D} sin(\phi) \tag{42}$$

The momentum theory is valid up to a critical value of the axial induction factor ($a < a_c$). For higher induction the wind turbine operating in the turbulent wake state. Thus, an empirical relationship between the thrust coefficient and the axial induction factor is needed. In this paper, Buhl and Marshall (2005) approach is used because it does not suffer from numerical discontinuity between the theoretical and empirical thrust coefficients and also it shows a good agreement with experimental

data. The proposed correction model is developed following the approach of Buhl and Marshall (2005). The thrust coefficient for the axial induction factor larger than a critical value ( $a > a_c = 0.4$) is given by Eq. (43).

$$C_T = \frac{8}{9} + \left(2\chi F - \frac{40}{9}\right)a + \left(\frac{50}{9} - 2\chi F\right)a^2 \tag{43}$$

It is clear that this equation is very similar to the original Buhl and Marshall (2005) correction. Equation 43 can be reduced to that of Buhl if relation the far wake expansion ratio $\chi = 2$.

## 4.2 Algorithm solution

Historically, the solution method for solving the BEM equations is based on taking the axial and tangential induction factors as the unknowns, then solve the fixed-point problem $(a, a') = f(a, a')$ using iterative method. This solution method is simple and fast but suffer from some convergence problems and does not always converge to the right solution (Maniaci , 2011) . Recently Ning (Ning , 2014) proposed a solution method that guaranteed the convergence which is based on reducing the two governing

equations of the BEM model to one residual function Eq. (44), then solve it by a root-finding algorithm. The inflow angle is the unknown and not the induction factors used as the unknown in the classical iterative algorithms.

$$f(\phi) = \frac{sin(\phi)}{1-a} - \frac{cos(\phi)}{(1+a')\lambda_r} \tag{44}$$





The momentum/empirical region corresponds to local inflow angles in the range $\phi \in C(]0, \pi[)$. In this case, there is two-equation for the axial induction factor, a criterion is needed to distinct the momentum region ($a < a_c$) and the empirical region

($a > a_c$). The criterion is given by ($\eta_0 = 2/3$). The axial and tangential induction factors can be computed by equating the thrust and torque in the MT and Kutta-Joukowski theory. The axial and tangential induction factors are given by Eq. (45) and Eq.(46), respectively.

$$a = \begin{cases} \frac{\eta}{1+\eta} & \eta \leq \eta_0 \\ \frac{\gamma_1 - \sqrt{\gamma_2}}{\gamma_3} & else \end{cases} \tag{45}$$

$$a' = \frac{\eta'}{1 - \eta'} \tag{46}$$

Where the parameters $\eta$, $\eta'$, $\gamma_1$, $\gamma_1$, and $\gamma_3$ are depending on the inflow angle. They are defined by:

$$\eta = \sigma C_{n3D} / (2\chi F sin^2(\phi)) \tag{47}$$

$$\eta' = \sigma C_{t3D} / (4F sin(\phi) cos(\phi)) \tag{48}$$

$$\gamma_1 = \chi F \eta - (10/9 - \chi/2F) \tag{49}$$

$$\gamma_2 = \chi F \eta - \chi/2F(4/3 - \chi/2F) \tag{50}$$

$$\gamma_3 = \chi F \eta - (25/9 - \chi F) \tag{51}$$

Once the induced velocities are converged, the performances of the wind turbine rotor can be computed using BET, since the drag force is accounted for when computing the performances of the wind turbine rotor. This is illustrated by the following algorithm.





---

**Algorithm 1** Solve $\phi^*$ for BEM model

**function** Root($x_l$,$x_h$,f)    ▷ Root finding algorithm
  $x^*$ where $f(x^*) = 0$ for $x_l \leq x \leq x_h$ and $f(x_l)f(x_h) < 0$
**end function**
**function** f($\phi$)    ▷ Residual function $f(\phi)$
  **if** $\eta < \eta_0$ **then**
   $a(\phi) = \frac{\eta}{1+\eta}$
  **else**
   $a(\phi) = (\gamma_1 - \sqrt{\gamma_2})/\gamma_3$
  **end if**
  $f(\phi) = \sin(\phi)/(1 - a(\phi)) - \cos(\phi)(1 - \eta'(\phi))/\lambda_r$
**end function**
**function** BEM    ▷ Main algorithm to solve BEM equations at a given blade radius
  $\epsilon = 10^{-6}$    ▷ Small value to avoid singualrity at $\phi = 0, \pm\pi$
  **if** $f(\epsilon)f(\pi/2) < 0$ **then**    ▷ The solution is within the range $]0, \pi/2[$
   $\phi^*$=Root($\epsilon, \pi/2, f(\phi)$)
  **else**
   **if** $f(\pi/2)f(\pi - \epsilon) < 0$ **then**    ▷ The solution is within the range $]\pi/2, \pi[$
    $\phi^*$=Root($\pi/2, \pi - \epsilon, f(\phi)$)
   **end if**
  **end if**
  $a = a(\phi)$
  $a' = a'(\phi)$
  $\phi = \text{atan}(\frac{1-a}{\lambda_r(1+a')})$
  $\alpha = \phi - \theta$
  $C_{n3D} = C_{l3D}(\alpha)\cos(\phi) + C_{d3D}(\alpha)\sin(\phi)$    ▷ Rotor performances accounting for drag force
  $C_{t3D} = C_{l3D}(\alpha)\sin(\phi) - C_{d3D}(\alpha)\cos(\phi)$
**end function**

---

## 5 Results and discussion

360   The improved BEM will be validated against experimental data of the NREL phase VI rotor, and it is also compared with the classical BEM. However, for briefness and relevance, the validation of the proposed improvements to the BEM method will be done by considering six different variations of the BEM method (Table 3), since three modifications to the classical BEM were proposed. First, an evaluation of the effect of the new far wake expansion model and K-J condition on the BEM solution using BEM-KJ-E-0 and BEM-KJ-E-0 models. Second, the validation of the improved stall delay model using BEM-S-0 and

365   BEM-S-1 models. Third, the spanwise distribution of the loading is evaluated and used to identify the wake expansion and stall delay effects on the prediction accuracy using BEM-S-E-0 and BEM-S-E-1 models. Last, An evaluation of the effect of wake expansion and KJ condition on the global loadings using BEM-KJ-E-0 and BEM-KJ-E-0 models.





**Table 3.** BEM model variations.

| BEM model variations | Stall delay model | Expansion model | K-J condition |
|---|---|---|---|
| BEM-KJ-E,0 | Improved Bak model | 1D far wake expansion | Including drag |
| BEM-KJ-E,1 | Improved Bak model | New far wake expansion | Excluding drag |
| BEM-S,0 | Bak model | New far wake expansion | Excluding drag |
| BEM-S,1 | Improved Bak model | New far wake expansion | Excluding drag |
| BEM-S-E ,0 | Bak model | 1D far wake expansion | Excluding drag |
| BEM-S-E,1 | Improved Bak model | New far wake expansion | Excluding drag |

## 5.1 3D BEM solution

The spanwise distribution of the converged angles of attack for different wind speed is shown in Fig. 5. The BEM-KJ-E,0 and
BEM-KJ-E,1 models are used and compared with the estimated AoA from experimental data given by Sant et al. (2006).

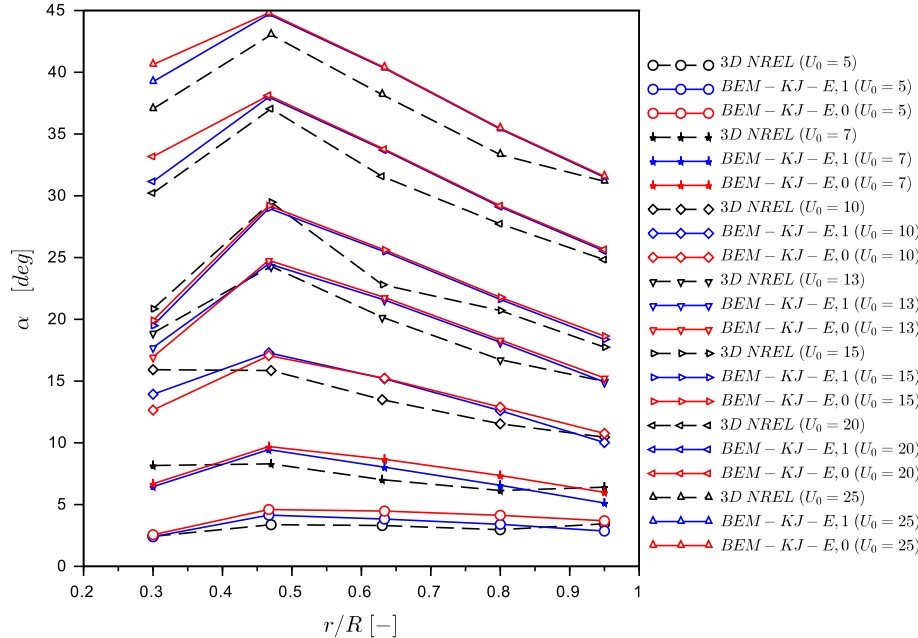

**Figure 5.** Spanwise distribution of angles of attack for different wind speed





At low wind speed ($U_0 \leq 10$), the drag force is negligible and the wake expansion effect on the AoA is important on the outboard sections of the blade. As a result of the new far wake expansion model (BEM-KJ-E,1 model), the converged angles of attack are in better agreement to experimental data compared using 1D far wake expansion model ( BEM-KJ-E,0model) that overpredict the AoA. However, at medium to high wind speed ($U_0 > 10$), the drag force becomes important and the expansion

effect is negligible (low tip speed). Additionally, the drag force is further increased at the hub region due to stall delay effect. Consequently, the converged AoA accounting for drag force (BEM-KJ-E,0 model), when solving BEM equations, are over-estimated near the hub region compared to the converged AoA computed by excluding drag force (BEM-KJ-E,1 model). However, for the full range of operating conditions, some inaccuracies of the converged AoA at the tip and hub are mainly due to the tip/hub loss model used. Nevertheless, both models are generally in good agreement with the AoA estimated from 3D

NREL phase VI experimental data.

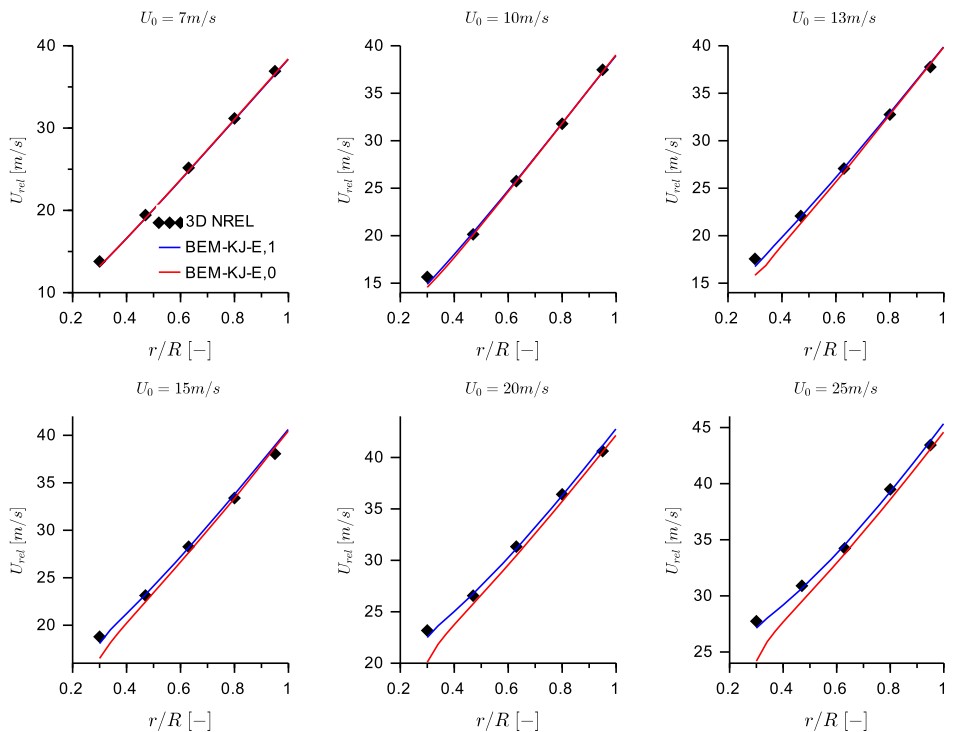

**Figure 6.** Spanwise distribution of relative velocity for different wind speed

For the NREL phase VI rotor, the axial and tangential induction factors are not available; however, the relative velocity can be estimated from the measured dynamic pressure. The spanwise distribution of the relative velocity, for different wind speed, is shown in Fig. 6. At low wind speed ($U_0 \leq 10$), both BEM-KJ-E,0 and BEM-KJ-E,1 models are accurately predicting the relative velocity of the NREL phase VI rotor, since the drag force is negligible for high tip speed and the new far wake





expansion model has no noticeable effect on the relative wind speed. As a result, the axial induction factor is under-estimated
in BEM-KJ-E,0 model since the AoA is over-estimated in BEM-KJ-E,0 model compared to BEM-KJ-E,1 model. However,
at medium to high wind speed ($U_0 > 10$), where the drag force becomes important, the relative velocity in the BEM-KJ-E,0
model diverge progressively toward the hub from the 3D NREL Phase VI experimental data, and the BEM-KJ-E,1 model is
accurately predicting the relative velocity for the full blade's span thanks to respecting KJ condition by excluding drag force.

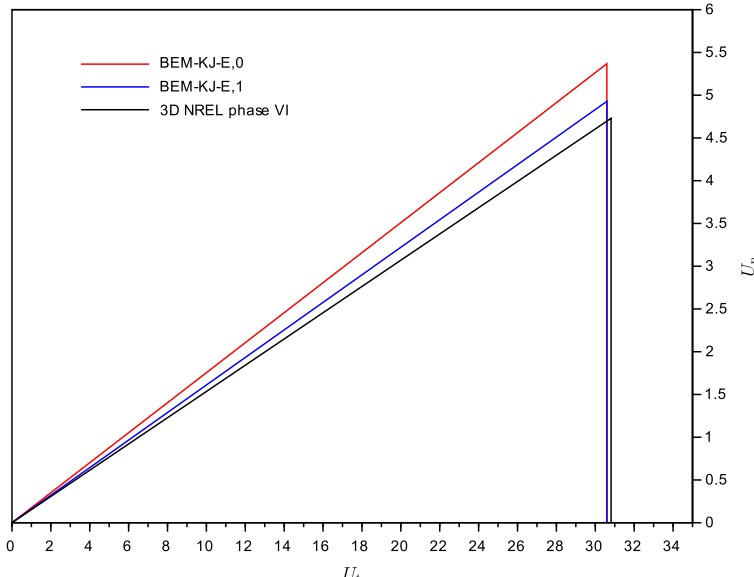

**Figure 7.** Velocity triangle at $r/R = 0.8$ and $U_0 = 7$

In order to further highlight the effect of the far wake expansion model on the induced velocities. The relative velocity
and inflow angles, at $r/R = 0.8$ and $U_0 = 7$, predicted using the BEM-KJ-E,0 and BEM-KJ-E,1 models along with the ones
estimated from experimental data are given in Figure 7. The axial induction factor predicted using the BEM-KJ-E,0 model is
under-predicted due to the use of the 1D far wake expansion model. However, the new far wake expansion model improves the
prediction of the axial induction factor, and it is in good agreement with the estimate axial induction from experimental data
as shown in Fig.7. The tangential induction factor is unchanged for both models since the torque coefficients in both models
are identical. The inflow angle predicted using the BEM-KJ-E,1 model is also improved compared to the BEM-KJ-E,0 model,
since the axial induction is under-estimated and the inflow angle is over-estimated. Thus, the new far wake expansion effect is
canceled out when computing the relative velocity as shown in Fig.6. This behavior remains valid for other wind speed cases.
    In order to further highlight the effect of excluding drag force on the induced velocities. The relative velocity and inflow
angles, at $r/R = 0.3$ and $U_0 = 20$, predicted using the BEM-KJ-E,0 and BEM-KJ-E,1 models along with the ones from ex-




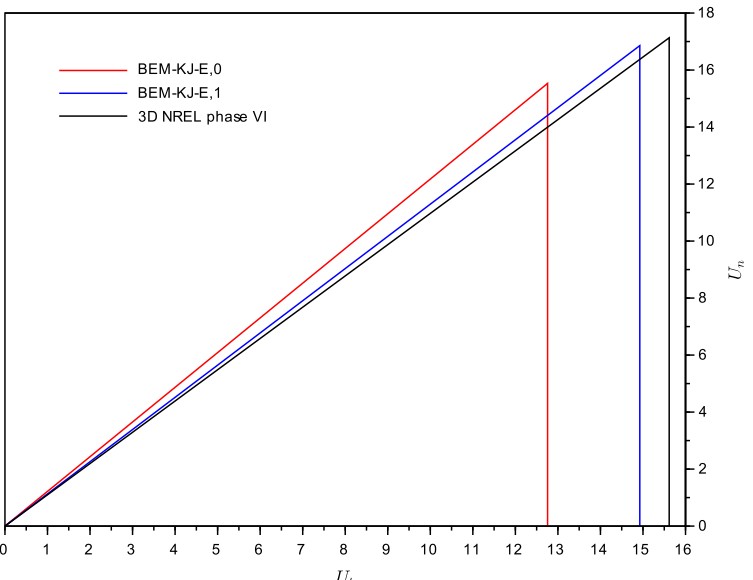

**Figure 8.** Velocity triangle at $r/R = 0.3$ and $U_0 = 20$

perimental data are given in Figure 8. Using the velocity triangle, it is clear that the axial induced velocity is over-estimated in the BEM-KJ-E,0 model, while the tangential induction factor is under-estimated because the drag force is considered when solving the BEM equations, which contradict the KJ theorem. However, When the KJ condition is respected, the axial and tangential induction factors are improved and correlate well with the ones estimated from experimental data. Additionally, the

inflow angle is slightly improved as shown in Fig. 5, and the relative velocity is significantly improved as shown in Fig. 6. This behavior remains valid for other wind speed cases.

Figure 9 shows the far wake expansion radius, at the free stream $U_0 = 7m/s$, predicted using the BEM-KJ-E,0 and BEM-KJ-E,1 models. Using the new far wake expansion model, The improved BEM predict accurately the wake expansion radius compared to the result of Large Eddy Simulation (LES) by Sezer-Uzol and Uzol (2013). However, the 1D MT expansion model

used in the BEM-KJ-E,0 model over-estimate the wake expansion radius, especially at high tip speed. It worth mentioning, that for low solidity wind turbines, the wake expansion radius converges after small distance downstream the rotor plan to an almost cylindrical wake. In the case of the NREL phase VI, the wake radius converges approximatively after one apparent helical pitch (Fig. 9). This was also confirmed by Madsen et al. (2010) showing that the axial velocity converges after just one diameter downstream of the rotor plane. Others like Sørensen and Kock (1995) and Zahle and Sørensen (2007) have also

shown similar behavior.





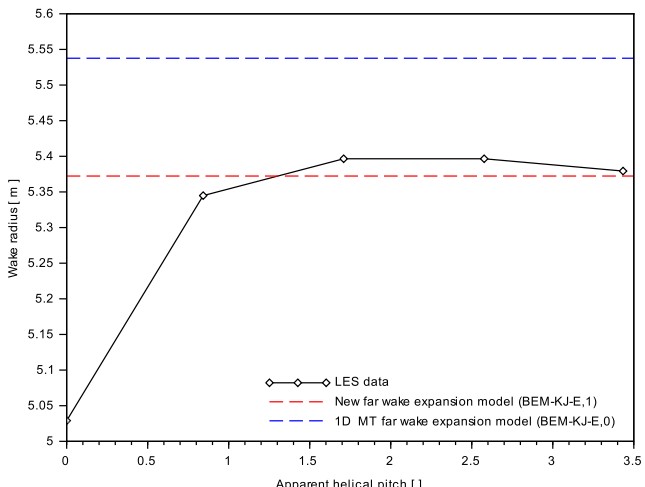

**Figure 9.** Wake expansion radius at $U_0 = 7$

### 5.2 3D stall delay model

Figure 10 shows the 3D sectional loading at five radial locations ($r/R = 30\%, 47\%, 63\%, \%80,$ and $95\%$). The BEM-S,0 and
BEM-S,1 models are used to validated the improved stall delay model. It can be seen that both stall delay models are identical
and predict relatively well the 3D behavior of the rotating blade, except in the separation area ($\alpha_0 < \alpha < \alpha_1$). In the separated
area, the improved Bak model predicts accurately the lift and drag coefficients, compared to Bak model that over-predict the
lift and drag due to integrating the pressure gradient over the whole chord. In fact, both stall delay models are only valid in
the separated area where the radial flow is dominant compared to the chordwise flow. As a result, the pressure gradient is
strongly dependent on the AoA, especially in the separation area, which was recently confirmed by Mauro et al. (2018) using
CFD simulation. At $r/R = 95\%$), the predicted loadings are in good agreement with experimental data, thanks to Shen tip loss
model that used to correct the lift and frag coefficients in both stall delay models.

In both stall delay models, the amplitude of the pressure gradient depends on external forces at a given spanwise position,
and the shape of the pressure gradient depends on the AoA and it is strongly empirical. Since it is taken from the 2D airfoil, then
corrected using an empirical quadratic function. The amplitude is generally in good agreement with experimental data (Fig.12
and Fig.13), however, some over-estimation and under-estimation of sectional loadings are due to the shape of the pressure
gradient that is strongly empirical and needs to be further studied. In fact, the shape of the pressure gradient can vary from
triangular shape to trapezoidal shape depending on the separation type: leading-edge separation or trailing edge separation.
Schreck and Robinson (2002) has analyzed the pressure data on the NREL Unsteady Aerodynamics Experiment and found



out that the 3D pressure coefficients have a trapezoidal as well as triangular shapes. The pressure shape variation was attributed to various mechanisms mediated by Coriolis and centrifugal forces and also to the presence of an energetic vortex structure.

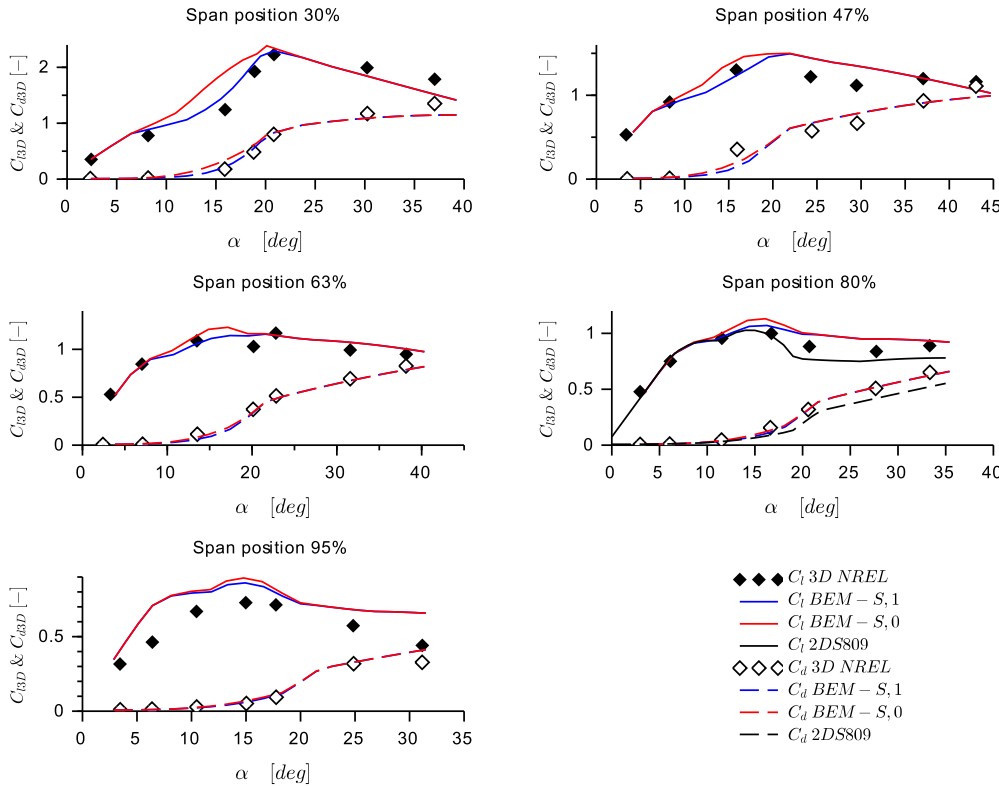

**Figure 10.** 3D sectional lift and drag coefficients

## 5.3 Local aerodynamic loadings

Figure 11 and 12 shows the normal and tangential force coefficient along the blade span for six different wind speed. The BEM-S-E,0 and BEM-S-E,1 model are used to evaluate the effect of far wake expansion and stall delay on the spanwise distribution of normal and tangential forces coefficients. The BEM-S-E,1 model agree very well with experimental data for the full range of operating conditions, compared to the BEM-S-E,0 model that over-predict the loading due to both using the 1D MT expansion model and the stall delay model of Bak. At hight tip speed ($U_0 \leq 10m/s$), the over-estimation of loadings by the BEM-S-E,0 is due to the use of 1D MT expansion model, that results in an over-estimation of the AoA, resulting in an over-estimation of normal force coefficient. At low tip speed ($U_0 \geq 10m/s$), the stall delay of Bak is responsible for the over-estimation of the spanwise distribution of both normal and tangential force coefficients at the hub region and at $U_0 = 7, 10m/s$, which is corresponding to the separation area. For the fully separated flow, both stall delay models are identical and they are in good





agreement with experimental data. A special case at $U_0 = 7m/s$ where the over-estimation of loading is due to both stall delay

model at inboard sections and wake expansion at the outboard sections.

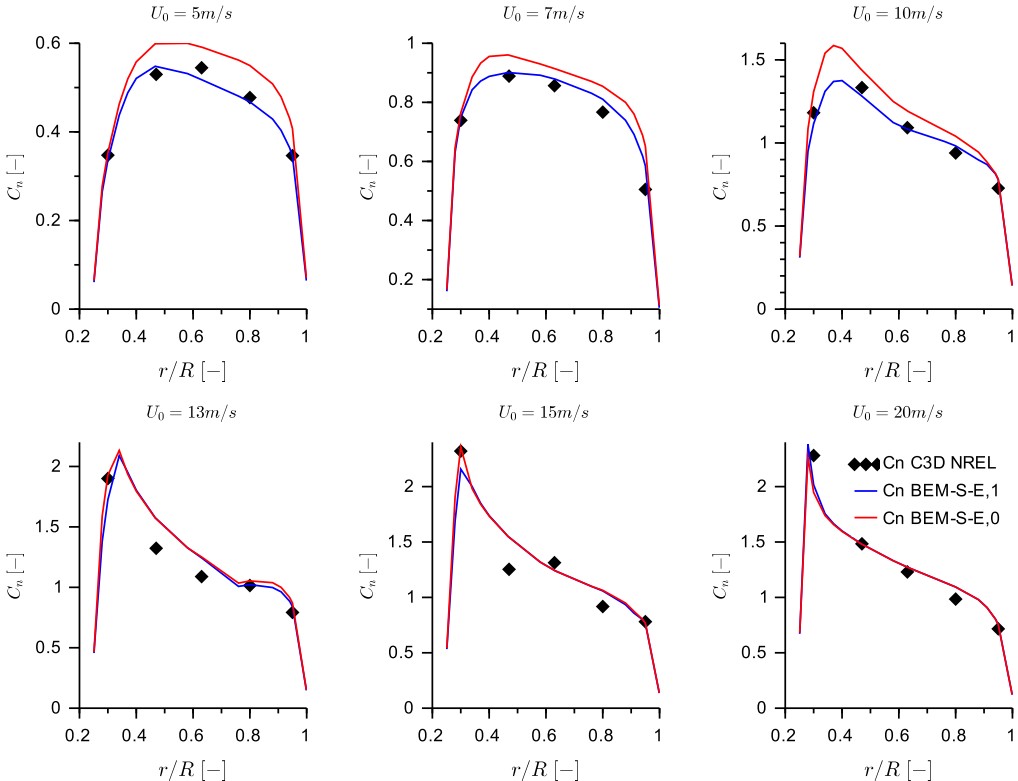

**Figure 11.** Spanwise distribution of the 3D normal force coefficient for different wind speed



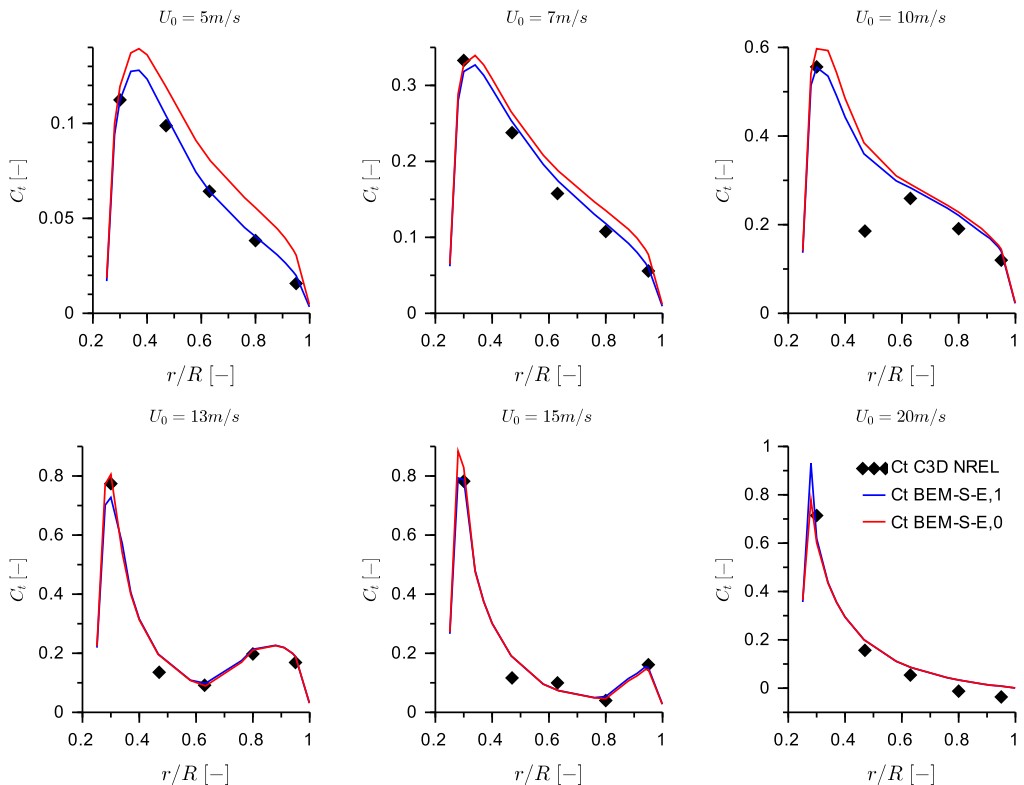

**Figure 12.** Spanwise distribution of the 3D tangential force coefficient for different wind speed

### 5.4 Global aerodynamic loadings

In order to see the far wake expansion and the KJ condition effect on the global performances, the classical BEM and the improved BEM are coupled with the improved stall delay model.

The power and thrust forces of the NREL Phase VI rotor for both BEM-KJ-E,0 and BEM-KJ-E,0 models are shown in Fig. 13. It can be seen that both models are predicting relatively well the power and thrust forces for the full range of operating conditions. However, at low wind speed (high tip speed) the power and thrust forces predicted using the BEM-KJ-E,0 model, are slightly overestimated compared to the BEM-KJ-E,0 model that account accurately for the far wake expansion effect. At medium to high wind speed (low tip speed), the power and thrust curves predicted using the BEM-KJ-E,0 model are under-

predicted compared to the BEM-KJ-E,0 model that predicts accurately the power and thrust force, thanks to respecting the Kutta-Joukowski condition.



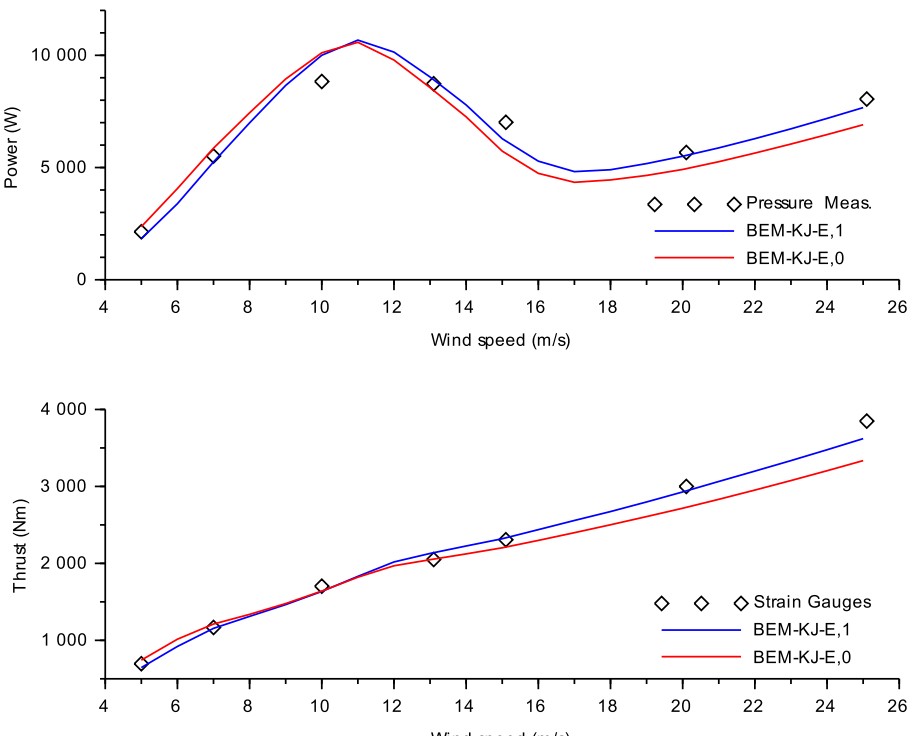

**Figure 13.** Trust and power for different wind speed

## 6 Conclusions

In this paper, a state of the art BEM model is developed for aerodynamic prediction of horizontal axis wind turbine rotor. The GMT is applied to low solidity rotors, then a new far wake expansion model is proposed. Three-dimensional effects on a rotating blade were estimated using an improved stall delay model based on Bak model. An improved solution method was given for the improved BEM model by respecting Kutta-Joukowski theorem and using the guaranteed convergence algorithm to solve the BEM equations. The improved BEM model is then compared against the classical BEM and the experimental data from the NREL phase VI rotor. The results show a remarkable agreement with experimental data. Additionally, the proposed improvements to the classical BEM requires virtually no additional computational time.

The new far wake expansion model was found to improve the spanwise loading and global loadings at high tip speed ratio, compared to the 1D MT expansion model that over-predict the spanwise loading and global loadings. As for the induced velocities, it was found that the axial induction factor was under-predicted using the 1D MT expansion model. While using the new far wake expansion model results in an accurate estimation of the axial induction factor. Additionally, The 1D MT





expansion model over-predict the far wake expansion radius, while the new far wake expansion model predicts accurately
the wake expansion radius. However, The new far wake expansion model can be further improved by taking into account the
additional velocity induced by the wake.

It was also found that accounting for drag force when solving BEM equations results in an important under-prediction of the
relative velocity in the hub region, while the converged AoA remains approximately unchanged. As for the induced velocities,
it was found that the axial induced velocity is over-estimation while the tangential induction factor is under-estimation in the
hub region. The thrust and torque forces were also under-predicted if the drag force is accounted for in BEM equations. In
contrast, excluding the drag force when computing the induced velocities, will result in an accurate prediction of the BEM
solution (axial induction, tangential induction, and inflow angle). Thus, local and global loadings were accurately predicted.

The three-dimensional effects on a rotating blade were accurately estimated using the improved stall delay model compared
to the original Bak model. However, Both models are still strongly empirical, especially the shape of the pressure gradient. As a
result, the chordwise pressure on a rotating blade needed to be further carefully analyzed in order to improve our understanding
of this complex phenomenon, thus accurately modeling the shape of the pressure gradient.

*Author contributions.* The developpement, implementation, and analysis of the improved BEM model was performed by YO, under the
supervision of AA. The preparation of the paper was performed by YO. AA provided assistance with the proof reading of the manuscript.

*Competing interests.* The authors declare that they have no conflict of interest

*Acknowledgements.* This work was supported by the NATO Science for peace grant program (SfP-982620) and the Moulay Ismail University
within the framework of its research support program 2018.



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
