# Peer review of "Accurate loads and velocities on low solidity wind turbines using an improved Blade Element Momentum model"

_Wind Energy Science, 2020_

## Referee Comment (RC1) · David Wood (Referee) · 16 Mar 2020

Blade element momentum (BEM) models have been used for many years for the aerodynamic analysis of wind turbines and propellers. Contrary to the statement on line 21, they were not introduced by Glauert (1935, reference in manuscript), who did however, develop them in a form that has been used for wind turbine analysis for nearly a century.  For example, Lock et al. (1925, reference below) gives a detailed account of the pre-Goldstein (1929) and pre-Glauert version of BEM.

Since the introduction of BEM, many second order corrections to it have been studied but these are not considered in the present manuscript.  They include the effects of finite blade number, e.g. Clifton-Smith (2009), Wood et al. (2016), Schmitz & Maniaci (2017), Wimhurst & Willden (2017, 2018), the nonlinearity of the governing equations, Wood & Okulov (2017).  Further, Limacher & Wood (2020) showed that the concerns of Goorjian (1972, reference in manuscript) over the role of the forces on the expanding streamtubes can be avoided easily.  Therefore the claim in the manuscript to present a "state of the art" BEM analysis is not valid.

The manuscript concentrates on two aspects of BEM: the effect of wake expansion and an extension of the model of Bak et al. (2006) for rotational effects on the blade element forces.  It is not demonstrated that these second order effects are more important than the others, but the reasonable agreement shown between the available measurements and the new model suggests that the contribution is worthwhile.

The axial induction factors, $a$ for the near-wake, and $b$ for the far-wake, are assumed to be independent of radius, in for example, Equations (3) and (4).  This is a major simplification which is not justified.  Constant $b$ in the far-wake gives a "Joukowsky" wake comprising a concentrated hub vortex of strength $N\Gamma$ where $\Gamma$ is the maximum bound circulation and $N$ is the number of blades, and $N$ helical vortices at the edge of the wake.  Then the relation between pitch and far-wake velocity is easily determined from the Kawada-Hardin equations (Kawada, 1936; Hardin, 1982) as

$$v = 1 - \frac{N\Gamma}{2\pi h}$$

Where $h$ is the vortex pitch, which contradicts Equation (12).  For a Joukowsky wake, Wood (2007) rediscovered the result of McCutchen (1985) that the rotational velocity contribution to the energy equation is cancelled by the contribution of the radial pressure gradient, and so can be ignored, contrary to the statement of its importance made several times in the manuscript.

It is also unlikely that the strength of the trailing vortices is set only by the lift on a blade element.  When the angular momentum equation, (6) in the manuscript, is balanced against the blade element forces, the element drag is involved, and, therefore, the rotational induction factor $a'$ is partly determined by the drag.  Since $a'$ is also the normalized circumferential velocity induced by the trailing helical vortices, the circulation of those vortices is also partly determined by the element drag.

The manuscript is poorly written in places.  For example, "plane" is often rendered as "plan" and there are many examples of poor expression which should be caught by a grammar checker.

Additional References

Clifton-Smith, M. J. (2009). Wind turbine blade optimisation with tip loss corrections. *Wind Engineering*, *33*(5), 477-496.

Goldstein, S. (1929). On the vortex theory of screw propellers. *Proceedings of the Royal Society of London. Series A, Containing Papers of a Mathematical and Physical Character*, *123*(792), 440-465.

Hardin, J. C. (1982). The velocity field induced by a helical vortex filament. *The Physics of Fluids*, *25*(11), 1949-1952.

Kawada, S. (1936). Induced velocity by helical vortices. *Journal of the Aeronautical Sciences*, *3*(3), 86-87.

Limacher, E. J., & Wood, D. H. Derivation of an Impulse Equation for Wind Turbine Thrust., *Wind Energy Science* https://www.wind-energ-sci-discuss.net/wes-2019-93/

Lock, C.N.H., Bateman, H., Townend, H.C.H. (1925) An extension of the vortex theory of airscrews with applications to airscrews of small pitch, including experimental results, ARC R & M, No. 1014, 1925.

McCutchen, C. W. (1985). A theorem on swirl loss in propeller wakes. Journal of Aircraft, 22(4), 344-346.

Schmitz, S., & Maniaci, D. C. (2017). Methodology to determine a tip-loss factor for highly loaded wind turbines. *AIAA Journal*, *55*(2), 341-351.

Wimshurst, A., & Willden, R. H. J. (2017). Analysis of a tip correction factor for horizontal axis turbines. *Wind Energy*, *20*(9), 1515-1528.

Wimshurst, A., & Willden, R. H. J. (2018). Computational observations of the tip loss mechanism experienced by horizontal axis rotors. *Wind Energy, 21*(7), 544-557.

Wood, D. H. (2007). Including swirl in the actuator disk analysis of wind turbines. Wind Engineering, 31(5), 317-323.

Wood, D. H., Okulov, V. L., & Bhattacharjee, D. (2016). Direct calculation of wind turbine tip loss. *Renewable Energy*, *95*, 269-276.

Wood, D. H., & Okulov, V. L. (2017). Nonlinear blade element-momentum analysis of Betz-Goldstein rotors. *Renewable Energy*, *107*, 542-549.

---

## Referee Comment (RC2) · Anonymous Referee #2 · 23 Jul 2020

Dear authors,

In my opinion, there are several issues with this article:

- The parameter $\xi$ is introduced wrongly just before Equation (11)

- There is a bracket too much in the denominator of Equation (14)

- In Equation (15) the term $\bar{h}$ in the square root in the denominator should be squared

- Any explanation why the empirical parameters $a_0$ and $b_0$ are both chosen as 1 for

this particular rotor (just after Equation (15)) is missing. Without any guidance on how to choose these parameters it is difficult to use the correction proposed in the paper.

- Equation (33) is not derived in the article and it is unclear where it comes from.

I leave it to the editor to decide how to proceed with the article and will provide a more detailed review if desired.

---

## Author Comment (AC1) · 23 Aug 2020

**Response to Referees for Wind Energy Science submission: Accurate loads and velocities on low solidity wind turbines using an improved Blade Element Momentum model by Yassine Ouakki and Abdelaziz Arbaoui**

Dear Referees,

We would like to thank the reviewers for their comments and guidance to improve the quality of our manuscript. The manuscript is submitted for publication in the special issue "Wind Energy Science Conference 2019" in Wind Energy Science journal. Comments from Reviewer 1 and Reviewer 2 are addressed here. The main changes to the manuscript are listed below:

- We rewrote the abstract to highlight the contribution of the article. The paper concentrate on two second-order effects, that are important for low solidity rotors.

- We updated the introduction by including additional references, proposed by reviewer 1, on other second-order corrections. The relative importance of these second-order effects is discussed for the case of a low solidity rotor.

- The method section is also updated by further clarifying the derivation of the proposed improvements to the classical BEM method. The hypothesis of excluding the drag force when computing the induced velocities is no longer considered in the manuscript.

- The results and discussion section is also updated for the changes made in the method section.

Again, thank you for your time and suggestions to improve our manuscript. Please find below a detailed answer to your comments.

On behalf of the authors, yours sincerely,

Attached:

- Response to Referee 1

- Response to Referee 2

- Marked-up manuscript version.

**Response to Referee 1**

Dear Professor D.H Wood,
We would like to thank you for your detailed review of our paper. Your recommendations and comments improved the paper considerably. Please, see below a detailed response to your comments:

**Comment 1:** Blade element momentum (BEM) models have been used for many years for the aerodynamic analysis of wind turbines and propellers. Contrary to the statement on line 21, they were not introduced by Glauert (1935, reference in manuscript), who did however, develop them in a form that has been used for wind turbine analysis for nearly a century. For example, Lock et al. (1925, reference below) gives a detailed account of the pre-Goldstein (1929) and pre-Glauert version of BEM.

**Reply 1:** Indeed, Glauert (1935) was not the first to introduce the BEM theory. The 1D momentum theory was applied to an actuator dick by Rankine (1865). Then, the blade element theory was developed by Froude (1898) and made the first combination of both theories for propeller analysis. Later, Glauert (1935) finalized the BEM method by describing a full BEM model including wake swirl, turbulent wake state based on experimental data by Lock et al. (1925), and tip loss concept by Prandtl (1226).
Lock et al. (1925) work is not available online, so we couldn't give a detailed account of the pre-Goldstein and the pre-Glauert version of BEM.

**Comment 2:** Since the introduction of BEM, many second order corrections to it have been studied but these are not considered in the present manuscript. They include the effects of finite blade number, e.g. Clifton-Smith (2009), Wood et al. (2016), Schmitz & Maniaci (2017), Wimhurst & Willden (2017, 2018), the nonlinearity of the governing equations, Wood & Okulov (2017). Further, Limacher& Wood (2020) showed that the concerns of Goorjian (1972, reference in manuscript) over the role of the forces on the expanding streamtubes can be avoided easily. Therefore the claim in the manuscript to present a state of the art BEM analysis is not valid.

**Reply 2:** We have updated the introduction to include recent papers on second-order corrections of the BEM method. First, the effect of finite blade number, including the nonlinearity of governing equations, are further discussed in the introduction. Second, the concerns of Goorjian (1972) can be avoided using the impulse formulation Limacher& Wood (2020). However, the pressure term due to wake expansion is still present in the GMT formulation, which is applied to a low solidity rotor in the present work.
In the conclusion section, the "state of the art BEM model" is replaced by "an improved BEM model".

**Comment 3:** The manuscript concentrates on two aspects of BEM: the effect of wake expansion and an extension of the model of Bak et al. (2006) for rotational effects on the blade element forces. It is not demonstrated that these second order effects are more important than the others, but the reasonable agreement shown between the available measurements and the new model suggests that the contribution is worthwhile.

**Reply 3:** The proposed improvements to the classical BEM model were a new far wake expansion model and an extension of Bak stall delay model.
The relative importance of second-order effects can be analyzed based on the rotor geometry and tip speed ratio. First, the GMT is applied to a low solidity rotor, as a result the pressure drop due to wale rotation can be neglected. However, the wake expansion effect is retained because the 1D wake expansion model over-estimate the loading and also gives an unrealistic

expansion at a high tip speed ratio. The effect of a finite number of blades is accurately accounted for using classical Glauert correction at a high tip speed ratio, where the new far wake expansion is validated. However, at low tip speed ratio, Glauert tip loss still over-estimate the loadings and can be improved using Wood et al. (2016) model. To analyze and improve the 1D MT expansion model at a high tip speed ratio, Glauert tip loss model can be used with confidence. The BET is corrected to two second-order effects: stall delay and tip loss. Both effects are considered in this work.

**Comment 4:** The axial induction factors, a for the near-wake, and b for the far-wake, are assumed to be independent of radius, in for example, Equations (3) and (4). This is a major simplification which is not justified. Constant b in the far-wake gives a Joukowsky wake comprising a concentrated hub vortex of strength N where is the maximum bound circulation and N is the number of blades, and N helical vortices at the edge of the wake. Then the relation between pitch and far-wake velocity is easily determined from the Kawada-Hardin equations (Kawada, 1936; Hardin, 1982) as

$$\nu = 1 - \frac{N\Gamma}{2\pi h} \tag{1}$$

Where h is the vortex pitch, which contradicts Equation (12). For a Joukowsky wake, Wood (2007) rediscovered the result of McCutchen (1985) that the rotational velocity contribution to the energy equation is cancelled by the contribution of the radial pressure gradient, and so can be ignored, contrary to the statement of its importance made several times in the manuscript.

**Reply 4:** In the case of a Joukowski rotor, the rotational velocity contribution to the energy equation is canceled by the contribution of the radial pressure gradient (wood 2007 and McCutchen 1985). However, in Joukowski model, the maximum power coefficient is always is greater than the Betz limit. This problem was discussed by Sorensen and van Kuik (2011). It was suggested that the large increase in Cp at small tip-speed ratios is the lack of the lateral pressure on the control volume. Even though, lateral pressure is small, neglecting it will result an unrealistic power coefficient at low tip speed ratio. A similar problem arises in Glauert 1D expansion model (b=2a), where the expansion area increases toward infinity with increasing axial induction.

In this work, The axial induction factors (a and b) are not assumed to be independent of radius in the GMT (sorensen 2016). So, the wake is not a Joukowski one, since the only additional assumption to GMT is neglecting the pressure drop due to wake rotation (low solidity). However, Neglecting the lateral force (wake expansion) will result in unrealistic expansion at high axial induction. Even though this term is small, it induces an unrealistic expansion (1D expansion model) and also an over-estimation of loading for the full blade span. Glauert model assumes simply that the axial velocity at the rotor disk ($U_n$) is a mean value of upstream velocity ($U_0$) and far wake axial velocity ($U_{nw}$). This relationship is independent of operating conditions and rotor geometry. To overcome this inconsistency, the far wake expansion will be modeled using dimensional analysis, and it will be used to refine the Glauert model for the effect of wake expansion for low solidity rotors.

The relation between the pitch and far-wake velocity (Equation bellow) is first determined geometrically, then approximated for small expansion case by neglecting the velocity deficit. So, the pitch becomes dependent on known rotor parameters such as blades number, tip speed ratio.

$$h = 2\pi r tan(\Phi) \tag{2}$$

Where, $\Phi$ is the angle between the vortex sheet and the rotor plane.

**Comment 5:**It is also unlikely that the strength of the trailing vortices is set only by the lift on a blade element. When the angular momentum equation, (6) in the manuscript, is balanced against the blade element forces, the element drag is involved, and, therefore, the rotational induction factor a is partly determined by the drag. Since a is also the normalized circumferential velocity induced by the trailing helical vortices, the circulation of those vortices is also partly determined by the element drag.

**Reply 5:**Yes, the Joukowski theorem is only valid for inviscid flow, where drag force is zero. However, a generalized JK theorem includes the drag force (reference below). So the drag force will be included in the algorithm solution of BEM equations.

Li, Juan, YiZhe Xu, and ZiNiu Wu. "Kutta-Joukowski force expression for viscous flow." SCIENCE CHINA Physics, Mechanics & Astronomy 58.2 (2015): 1-5.

**Comment 6:**The manuscript is poorly written in places. For example, plane is often rendered as plan and there are many examples of poor expression which should be caught by a grammar checker.

**Reply 6:**Grammar was checked and gramatical correction where made.

**Response to Referee 2**

Dear Reviewer,

Thank you for your preliminary review of our paper. Your comments improved the paper considerably. Please see below a detailed response to your comments:

**Comment 1:** The parameter $\xi$ is introduced wrongly just before Equation (11).
**Reply 1:** Corrected.

**Comment 2:** There is a bracket too much in the denominator of Equation (14)
**Reply 2:** Corrected.

**Comment 3:** In Equation (15) the term $\bar{h}$ in the square root in the denominator should be squared
**Reply 3:** Corrected.

**Comment 4:** Any explanation why the empirical parameters $a_0$ and $b_0$ are both chosen as 1 for this particular rotor (just after Equation (15)) is missing. Without any guidance on how to choose these parameters it is difficult to use the correction proposed in the paper.
**Reply 4:** In order to estimate the wake expansion ratio $\xi$, we applied the Buckingham $\pi$ theorem to derive a physically meaningful equation (Equation 15) involving a certain number of dimensionless numbers (table 2). These dimensionless numbers can be further regrouped to form the apparent helical pitch $\bar{h}$.
By applying the $\pi$ theorem, two constant parameters $a_0$ and $b_0$ need to be estimated. First, using the limiting case ($\lambda$ converge to 0), the parameter $a_0$ is found to be equal to unity. The parameter $b_0$ has been estimated from experimental data of the NREL phase VI. Only data at one tip speed ratio is enough to estimate $b_0$. However, to cover a broader range of tip speed ratio, we used the data of the NREL phase VI at $U_0 = 5m/s, U_0 = 7m/s$, corresponding to a tip speed of 7.58 and 5.41 respectively. Comparing computed and measured distributions of normal and tangential forces, it was found that $b_0 \approx 1$ gives the best curve fit.

**Comment 5:** Equation (33) is not derived in the article and it is unclear where it comes from.
**Reply 5:** Bak et al. assumed that the chordwise flow is zero in the separated area, which is the same assumption made by Corten and Lindenburg to describe the centrifugal pumping mechanism due to rotation. Thus, the radial flow is assumed to be dominant compared to the chordwise flow. The centrifugal pumping mechanism is described as follows, the centrifugal force induces a radial flow, which in turn induce a chordwise Coriolis force that delays separation and amplifies the pressure in the separated area.
The normal force gradient was computed by integrating the pressure gradient over the full chord (Eq.29 in the manuscript), which contradicts the initial assumption that the rotational effect is present only in the separated area over the airfoil. Thus we proposed in this paper to estimate the normal force gradient by integrating the pressure gradient in the separated area only (Eq.33 in the manuscript). Additionally, assuming that the pressure gradient induced by the radial flow in the separated area mainly affects the normal force, thus the tangential force gradient can be neglected. The proposed improvement is consistent with the centrifugal pumping mechanism and the normal force gradient becomes dependent on the separation factor, which is a measure of the separated area on the airfoil.

**Accurate loads and velocities on low solidity wind turbines using an improved Blade Element Momentum model**

Yassine Ouakki[1] and Abdelaziz Arbaoui[1]

[1]INSCM team, LM2I, National School of Arts and Crafts (ENSAM), Moulay Ismail University, BP 4024 Meknes, Morocco

**Correspondence:** Yassine Ouakki (yassineouakki@gmail.com)

**Abstract.**

The accurate prediction of loadings and velocities on a wind turbine blades is essential for the design and optimization of wind turbines rotors. However, the classical BEM still suffer from an inaccurate prediction of induced velocities and loadings, even if the classical  corrections like stall delay effect and tip loss correction are used. For low solidity rotors, the  far wake expansion is generally considered to be small, even at high tip speed ratio. In contrast, the 1D far wake expansion  model gives an unrealistic expansion of the wake at a high tip speed ratio. This inconsistency is observed in Glauert model, where the far wake axial induction is  taken as twice the axial induction in the rotor plane  . This unrealistic behavior of the 1D far wake expansion can be avoided by estimating the expansion ratio using dimensional analysis. Due to the complex nature of the flow around a rotating blade, the accurate estimation of 3D effects is still challenging, since most stall delay models still often tend to under-predict or over-predict the loadings near the root region. ~~As for the solution method for the classical BEM equation, the induced velocities are computed accounting for the drag force. However, according to the Kutta-Joukowski theorem, the induced velocities on a blade element are only created by lift force. Accounting for drag force when solving the BEM will result in an over-estimation of the axial induction factor, while the tangential induction factor is under-estimated.to take into account the radial flow effectAn improved solution method for the BEM equations respecting the Kutta-Joukowski theorem is proposed.~~ The improved BEM model is used to estimate the aerodynamic loads and velocities on the National Renewable Energy Laboratory Phase VI rotor blades. The results of this study show that the proposed BEM model gives an accurate prediction of the loads and velocities compared to the classical BEM model.

**1 Introduction**

Thanks to its simplicity and robustness, the blade element momentum (BEM) theory has been widely used in the wind industry for the design and optimization of wind turbine rotors (Tangler, 2002).  The BEM theory is a combination of  momentum theory (MT) and  blade element theory (BET). The 1D MT,

first proposed by Rankine (1865), is describing the process of energy extraction by the wind turbine rotor using the conservation laws of fluid mechanics. The BET, developed by Froude (1878) , is describing the local loadings and velocities  computations on a blade element using airfoil theory. ~~Some assumptions in the BEM theory are corrected to make the prediction more realistic. The assumption of an infinite number of blades in the 1D MT is corrected using the Prandtl (1921) tip and hub loss model. The three-dimensional effects (3D stall delay) are also taken into account by correcting the 2D aerodynamic coefficients in the BET. However, The BEM model still suffers from its inaccurate prediction of induced velocities and the non-dimensional loading.~~

Later, Glauert (1935) finalized the BEM method by describing a full BEM method including wake swirl, turbulent wake state based on experimental data by Lock (1925) and tip loss concept by Prandtl (1921). Since then many engineering corrections were proposed to refine some assumptions in both theories. The 1D MT equations are  derived from the general momentum theory (GMT) with additional assumptions. The first simplification was neglecting all nonlinear tangential velocity terms; as a result, the pressure drop due to wake rotation was neglected. The second simplification was using the GMT equation in differential form, which has been proven to be wrong by Goorjian (1972). Consequently, the axial induction in the far wake is taken as twice the induction at the rotor disc. Very recently, Limacher and Wood (2020) derived an impulse formulation having the advantage of avoiding the pressure term arising from the expansion of the streamtube, thus the concern of Goorjian (1972) can be avoided. However, the GMT formulation still accounts for the pressure term due to wake expansion.

Joukowski (1912) and Sharpe (2004) considered the effect of the pressure drop due to wake rotation, they found out that the rotor power coefficient increases at low tip speed ratio. Wood (2007) confirmed the finding of McCutchen (1985) that the rotational velocity contribution to the energy equation is canceled by the contribution of the radial pressure gradient. However, in Joukowski model, the maximum power coefficient is always exceeding the Betz limit and that it increases toward infinity at a low tip-speed ratio. This problem was discussed by Sørensen and van Kuik (2011) . It was suggested that the large increase in power coefficient, at small tip-speed ratios, is due to the lack of the lateral pressure on the control volume. Neglecting the lateral pressure, even if the lateral pressure is low, will result in an unrealistic power coefficient at a low top speed ratio. A similar problem arises in Glauert 1D expansion model, where the expansion area increases toward infinity with increasing tip-speed ratio. The influence of the pressure drop due to wake rotation is important for slow running rotors when the tip-speed-ratio is small (Vaz et al., 2011) . In contrast, For modern wind turbine rotors operating with  high rotational speed and low torque (low solidity, typically 7% or less), the rotational kinetic energy in the wake will be small (Gupta and Leishman , 2005). As a result, the pressure drop due to wake rotation can be neglected. But, the use of an erroneous differential form of the GMT, will result in the cancelation of wake expansion by the pressure drop due to wake rotation. The power loss from wake rotation at a low tip speed ratio will be almost canceled by the increased mass flow through the rotor (De Vries (1979), Sharpe (2004), and Xiros and Xiros (2007)). However, at the level of spanwise loading, the wake expansion and pressure drop due to wake rotation do not cancel each other. In fact, The spanwise loading will be reduced at the tip region due to the presence of radial flow, and it will be augmented at the hub region due to the pressure drop due to wake rotation ( Mast et al. (2004), Dossing et al. (2012), and Madsen et al. (2010)). Recently, Sun et al. (2016) proposed

[revised manuscript text omitted]

 The aerodynamic losses are present in both BET and MT. The MT assumes that the rotor has an infinite number of blades. In reality, the rotor has a finite number of blades and the blade has a finite span. Prandtl (1921) has shown that ~~in the case of a non-swept rotor blade , the relative induced velocity on the airfoil is a result of the circulation around it. Thus, it is perpendicular to the local relative flow. Wilson and Lissaman (1974) pointed out that the drag should be excluded from BEM equations because the drag-based velocity deficit is only a characteristic of the wake and it does not contribute to the induced velocity at the rotor disc~~ the blade circulation tend to zero at the blade's tip. Later, Glauert (1935) proposed a simple approximation of Prandtl (1921) tip loss, which is commonly used in BEM codes. Glauert (1935) tip loss corrects the axial and tangential induction factors. Wilson and Lissaman (1974) and De Vries (1979) refined it by correcting the mass flux. De Vries (1979) correction was found to be the most suitable for blade optimization. However, the power coefficient improves about 1% compared to the BEM without tip loss (Clifton-Smith , 2009). Recently,  Wood et al. (2016) proposed three methods for direct calculation of tip  loss, using helical vortex theory, that improve the prediction for all tip speeds. Especially, at low tip speed where the classical Glauert correction is inaccurate. Later, Wood and Okulov (2017) analyzed the nonlinear terms in the thrust and torque coefficient due to finite number of blades . Nonlinear terms were found to be important in the tip region at a low tip speed ratio. Glauert (1935) tip loss model is adopted in this work because it is accurate at a high tip speed ratio, where the expansion effect is important. The blade element theory neglect the span-wise dimension effect by using 2D airfoil data to compute the loads near the tip, however, the loads should tend to zero at the tip allowing to pressure equalization Shen et al. (2005). The tip loss mechanism was studied by Wimshurst and Willden (2018), and it was attributed to the effect of the vorticity shedding from the outboard blade sections. Shen et al. (2005) proposed a modified tip loss of Glauert to take into account this kind of aerodynamic loss. Recently, Shen tip loss correction was re-calibrated in axial and tangential directions separately, using Mexico data (Wimshurst and Willden , 2017). Thus, improving the tangential loading

and power predictions. Schmitz and Maniaci (2017) adopted Shen et al. (2005) approach to correct the Glauert tip loss for the effect of tip roll-up vortex and wake expansion in the tip region.

This manuscript is organized as follows. In Section 2, the GMT is applied to low solidity wind turbine rotors, then a new far wake expansion model is proposed using dimensional analysis. In Section 3, the blade element model is described, then an improved stall delay model is proposed. In Section 4,  The improved BEM equations are solved using the guaranteed convergence algorithm. In Section 5, A detailed validation of the proposed BEM model is performed based on the experimental results of the NREL Phase VI rotor (Hand et al , 2001). At last, the conclusion and further improvements of the BEM model will be given.

**2 Improved momentum theory**

**2.1 General momentum theory applied to low solidity wind turbines**

[revised manuscript text omitted]

The lateral force component, $dT_{p,Side}$ is difficult to determine and can be found using CFD, however, this term is generally considered to be small (Sørensen , 2016). Thus, the annular elements are independent of each other, and the far wake axial induction is taken twice the axial induction at the rotor disc (b=2a), corresponding to Glauert 1D MT model. However, neglecting the lateral force component will result in an unrealistic wake expansion area in the 1D far wake expansion model (Eq. (5)) at higher axial induction (see Fig. 2), which contradict the initial assumption of annular independence. Consequently, Glauert model is only consistent at a very low axial induction factor, where both the governing equation (Eq. (3)) and continuity equation (Eq. (5)) respect the annular independence and small expansion conditions. Glauert model assumes simply that the axial velocity at the rotor disk ($U_n$) is a mean value of upstream velocity ($U_0$) and far wake axial velocity ($U_{nw}$). This relationship is independent of operating conditions and rotor geometry. In order to ensure the small expansion for higher axial induction, the expansion ratio ($\chi = b/a$) needs to be accurately estimated by taking into account the operating conditions and rotor geometry.

$$\frac{r_w}{r} = \sqrt{\frac{1-a}{1-2a}} \qquad (5)$$

The thrust coefficient becomes similar to 1D MT except for the expansion effect that is parameterized by the far wake expansion ratio ($\chi = b/a$) . the expansion ratio will be accurately estimated instead of taking $b = 2a$ as the 1D MT. The hypothesis of an infinite number of blades is corrected using the Prandtl tip and hub loss model (Prandtl, 1921) . The thrust and torque are given by Eq. (6) and Eq. (7), respectively.

$$C_T = 2\chi a F(1-a) \qquad (6)$$

$$C_Q = 4a' F(1-a)\lambda_r \qquad (7)$$

[Figure]

**Figure 2.** 1D MT Far wake expansion radius as a function of axial induction factor

[revised manuscript text omitted]

$$\chi(r) = 1 + a_0 \left( \frac{\bar{h}}{\sqrt{1 + \bar{h}}} \frac{\bar{h}}{\sqrt{1 + \bar{h}^2}} \right)^{b_0} \tag{15}$$

Where $a_0$ and $b_0$ are empirical parameters.  First, using the limiting case $(\lambda \to 0)$, the parameter $a_0$ is found to be always equal to unity $(a_0 = 1)$. The parameter $b_0$ has been estimated from experimental data of the NREL phase VI. Only data at one tip speed ratio is enough to estimate $b_0$. However, in order to cover a broader range of tip speed ratio, we used the data of the NREL phase VI at $U_0 = 5m/s, U_0 = 7m/s$, corresponding to a tip speed of 7.58 and 5.41 respectively. Comparing computed and measured distributions of normal and tangential forces, it was found that $b_0 \approx 1$ gives the best curve fit. Once the far wake expansion ratio is found, the new far wake expansion radius can be computed using Eq.16 as follows:

$$\frac{r_w}{r}(r) = \sqrt{\frac{1 - a(r)}{1 - a(r)\chi(r)}} \tag{16}$$

**3 Blade element model accounting for stall delay effect**

**3.1 Blade element theory**

The blade element theory assumes that the blades are made up of a number of blade elements arranged in the spanwise direction, so the airfoil theory can be used to compute the local loads and velocities acting on each blade element. The BET assumes that the blade elements are aerodynamically independent and do not have any interference between them. The loads can be obtained from the 2D lift, drag, and moment coefficients of the airfoil at any radial position. The inflow is known at the blade and a lifting-line assumption is used (Branlard , 2017). As a result of these assumptions, the global loads on a blade can be integrated over the total blade span incorporating the velocity terms, to obtain the thrust, the torque, and power developed by the blade. This is further multiplied by the number of blades to get the total rotor thrust, torque, and power.

[Figure]

**Figure 3.** Velocity triangle and resulting aerodynamic  forces applied to an airfoil.

A blade element is an infinitesimal fraction dr of the blade radius R at a radial position r, so it can be considered as an airfoil. An airfoil is defined with by its chord c(r), its twist $\theta(r)$ and its shape. The relative wind applied to a blade element, noted $U_{rel}(r)$, is decomposed into a normal component $U_n(r)$ and a tangential component $U_t(r)$ to the rotor plane. Three characteristic angles defined by the velocity axis and the chord axis: the flow angle $\phi(r)$ between the tangential and relative velocity and it is assumed to be known, the airfoil twist $\theta(r)$ about to the rotor plane, and the angle of attack $\alpha(r)$ between the relative velocity and the chord axis and it is related to aerodynamic loads. Since the lifting-line assumption is used, the velocity triangle can be defined in terms of the axial and tangential inductions factors $a$ and $a'$ (Eq. 17 and Eq.18). The relationship between these angles and the velocity components are given in Equations 19 and 20. The blade element, velocity triangle, and aerodynamic forces are shown in Figure 3.

$$U_n = U_0(1 - a) \tag{17}$$

$$U_t = \Omega r(1 + a') \tag{18}$$

$$tan(\phi) = \frac{U_n}{U_t} \tag{19}$$

$$\alpha = \phi - \theta \tag{20}$$

The lift and drag forces per unit of length applied on the blade element of the surface are defined as follows:

$$L = 1/2\rho c U_{rel}^2 C_{l2D} \tag{21}$$

$$D = 1/2\rho c U_{rel}^2 C_{d2D} \tag{22}$$

[revised manuscript text omitted]

**Algorithm 1** Solve $\phi^*$ for BEM model

```
function Root(x_l,x_h,f)                                              ▷ Root finding algorithm
    x* where f(x*) = 0 for x_l ≤ x ≤ x_h and f(x_l)f(x_h) < 0
end function
function f(φ)                                                        ▷ Residual function f(φ)
    if η < η_0 then
        a(φ) = η/(1+η)
    else
        a(φ) = (γ_1 − √γ_2)/γ_3
    end if
    f(φ) = sin(φ)/(1 − a(φ)) − cos(φ)(1 − η'(φ))/λ_r
end function
function BEM                              ▷ Main algorithm to solve BEM equations at a given blade radius
    ε = 10^{−6}                                   ▷ Small value to avoid singualrity at φ = 0, ±π
    if f(ε)f(π/2) < 0 then                            ▷ The solution is within the range ]0, π/2[
        φ*=Root(ε, π/2, f(φ))
    else
        if f(π/2)f(π − ε) < 0 then                   ▷ The solution is within the range ]π/2, π[
            φ*=Root(π/2, π − ε, f(φ))
        end if
    end if
    a = a(φ)
    a' = a'(φ)
    φ = atan(  (1−a) / (λ_r(1+a'))  )
    α = φ − θ
    C_{n3D} = C_{l3D}(α)cos(φ) + C_{d3D}(α)sin(φ)
    C_{t3D} = C_{l3D}(α)sin(φ) − C_{d3D}(α)cos(φ)
end function
```

**5.1 3D BEM solution**

430  The spanwise distribution of the converged  angle of attack for different wind speed is shown in Fig. 4. The  C-BEM and N-BEM models are used and compared with the estimated AoA from experimental data given by Sant et al. (2006).

At  low wind speed ($\cancel{U_0 \leq 10}$ $U_0 < 10$), the drag force is negligible and the wake expansion effect on the AoA is important on the outboard sections of the blade. As a result of the new far wake expansion model ( N-BEM), the

435  converged angles of attack are in better agreement to experimental data compared  to the 1D far wake expansion model ( C-BEM) that overpredict the AoA. However, at medium to high wind speed ($\cancel{U_0 > 10}$ $U_0 \geq 10$), the  stall delay effect becomes important and the expansion effect is negligible . The improved Bak model (N-BEM) gives better agreement to experimental data, and the original Bak model underestimates the AoA at the hub region

440   when the flow is not yet fully separated. Once the flow becomes fully separated, both N-BEM and C-BEM models are identical. However, for the full range of operating conditions, some

[Figure]

**Figure 4.** Spanwise distribution of  angle of attack for different wind speed

inaccuracies of the converged AoA at the tip and hub are mainly due to the tip/hub loss model used. Nevertheless, both models are generally in good agreement with the AoA estimated from 3D NREL phase VI experimental data.

445    For the NREL phase VI rotor, the axial and tangential induction factors are not available; however, the relative velocity can be estimated from the measured dynamic pressure. The spanwise distribution of the relative velocity, for different wind speed, is shown in Fig. 5. At low wind speed ($U_0 < 10$), both  N-BEM and C-BEM models are accurately predicting the relative velocity of the NREL phase VI rotor  since the new far wake expansion model has no noticeable effect on the relative wind speed. As a result, the axial induction

450    factor is under-estimated in  the C-BEM model since the AoA is over-estimated in  the C-BEM model. The N-BEM model accurately predicts the axial induction factor at a high tip speed ratio, thanks to the new far wake expansion model (Figure 6). However, at medium to high wind speed ( $U_0 \geq 10$), the relative velocity in  both C-BEM and N-BEM models diverge progressively toward the hub from the 3D NREL Phase VI experimental data,

455     because of the over-estimation of AoA at low tip speed ratio.

[Figure]

**Figure 5.** Spanwise distribution of relative velocity for different wind speed

In order to further highlight the effect of the far wake expansion model on the induced velocities. The relative velocity and inflow angles, at $r/R = 0.8$ and $U_0 = 7$, predicted using the  C-BEM and N-BEM models along with the ones estimated from experimental data are given in Figure 6. The axial induction factor  is  under-predicted using the C-BEM model, due to the use of the 1D far wake expansion model. However, the new far wake expansion model (N-BEM) improves the prediction of the axial induction factor, and it is in good agreement with the  estimated axial induction from experimental data as shown in Fig.6. The tangential induction factor is unchanged for both models since the torque coefficients in both models are identical. The inflow angle predicted using the  N-BEM model is also improved compared to the  C-BEM model. Using the C-BEM, the axial induction is under-estimated and the inflow angle is over-estimated. As a result, the new far wake expansion effect is canceled out when computing the relative velocity as shown in Fig.5. This behavior remains valid for other wind speed cases.

[Figure]

**Figure 6.** Velocity triangle at $r/R = 0.8$ and $U_0 = 7$

[revised manuscript text omitted]

Mauro S, Lanzafame R, Messina M, Brusca S.: A Detailed Analysis of the Centrifugal Pumping Phenomenon in HAWTs Through the Use of CFD Models. Colloquium on Research and Innovation on Wind Energy on Exploitation in Urban Environment Colloquium. Springer, Cham, 2018.

Madsen HA, Bak C, Døssing M, Mikkelsen R, Øye S.: Validation and modification of the blade element momentum theory based on com-
610 parisons with actuator disc simulations. Wind Energy: An International Journal for Progress and Applications in Wind Power Conversion Technology 13.4, 373-389, 2010.

Maniaci D.: An investigation of WT-perf convergence issues. 49th AIAA Aerospace Sciences Meeting including the New Horizons Forum and Aerospace Exposition. 2011.

Mast EH, Vermeer LJ, Van Bussel GJ.: Estimation of the circulation distribution on a rotor blade from detailed near wake velocities. Wind
615 Energy: An International Journal for Progress and Applications in Wind Power Conversion Technology 7.3, 189-209, 2004.

McCutchen, C. W.: A theorem on swirl loss in propeller wakes. Journal of Aircraft, 22(4), 344-346, 1985.

Micallef D, van Bussel G, Ferreira CS, Sant T.: An investigation of radial velocities for a horizontal axis wind turbine in axial and yawed flows. Wind Energy 16.4, 529-544, 2013.

Ning, SA.: A simple solution method for the blade element momentum equations with guaranteed convergence. Wind Energy 17.9, 1327-
620 1345, 2014.

Okulov VL, Sørensen JN, Wood DH.: The rotor theories by Professor Joukowsky: Vortex theories. Progress in aerospace sciences 73, 19-46, 2015.

Prandtl, L.: Applications of modern hydrodynamics to aeronautics, (Tech- nical report No. 116). National Advisory Committee for Aeronautics (NACA), 1921.

625 Pratumnopharat P, Leung PS.: Validation of various windmill brake state models used by blade element momentum calculation. Renewable energy 36.11, 3222-3227, 2011.

Rankine, W. J. M.: On the Mechanical Principles of the Action of Propellers. Transactions, Institute of Naval Architects, Vol. 6: pp. 13-30.

Sant T, van Kuik G, Van Bussel GJ.: Estimating the angle of attack from blade pressure measurements on the NREL phase VI rotor using a free wake vortex model: axial conditions. Wind Energy: An International Journal for Progress and Applications in Wind Power Conversion
630 Technology 9.6, 549-577, 2006.

Schreck S, Robinson M.: Rotational augmentation of horizontal axis wind turbine blade aerodynamic response. Wind Energy: An International Journal for Progress and Applications in Wind Power Conversion Technology 5.2-3, 133-150 2002.

Sezer-Uzol N, Uzol O.: Effect of steady and transient wind shear on the wake structure and performance of a horizontal axis wind turbine rotor. Wind Energy 16.1, 1-17, 2013.

635 Sharpe D.J.: A general momentum theory applied to an energy-extracting actuator disc, Wind Energy 7 (3), 2004.

Shen WZ, Mikkelsen R, Sørensen JN, Bak C.: Tip loss corrections for wind turbine computations. Wind Energy: An International Journal for Progress and Applications in Wind Power Conversion Technology 8.4, 457-475, 2005.

Schmitz, S., Maniaci, D. C. : Methodology to determine a tip-loss factor for highly loaded wind turbines. AIAA Journal, 55(2), 341-351. 2017.

640 Snel H, Houwink R, Bosschers J.: Sectional prediction of lift coefficients on rotating wind turbine blades in stall. ECN-C-93-052, ECN, Petten, 1993.

Sørensen JN, Kock CW.: A model for unsteady rotor aerodynamics. Journal of wind engineering and industrial aerodynamics 58.3, 259-275, 1995.

Sørensen JN, Shen WZ, Munduate X.: Analysis of wake states by a full-field actuator disc model. Wind Energy: An International Journal for

645 Progress and Applications in Wind Power Conversion Technology 1.2, 73-88, 1998.

Sørensen, JN.: General momentum theory for horizontal axis wind turbines. Vol. 4. Switzerland: Springer, 2016.

Sørensen, Jens N., and Gijs AM van Kuik. : General momentum theory for wind turbines at low tip speed ratios. Wind Energy 14.7 (2011): 821-839.

Spera DA., Wilson RE.: Wind turbine technology : fundamental concepts of wind turbine engineering. Second Edition, 2009

650 Sun Z, Chen J, Shen WZ, Zhu WJ.: Improved blade element momentum theory for wind turbine aerodynamic computations. Renewable energy 96, 824-831, 2016.

Tangler James L. : The Evolution of Rotor and Blade Design, 2000

TanglerJ.: The nebulous art of using wind-tunnel airfoil data for predicting rotor performance, ASME 2002 Wind Energy Symposium. American Society of Mechanical Engineers, 2002.

655 Van Kuik GA.: Joukowsky actuator disc momentum theory. Wind Energy Science 2.1, 307, 2017.

Van Kuik GA, Lignarolo LE.: Potential flow solutions for energy extracting actuator disc flows. Wind Energy 19.8, 1391-1406, 2016.

Vaz JR, Pinho JT, Mesquita AL.: An extension of BEM method applied to horizontal-axis wind turbine design. Renewable Energy 36.6, 1734-1740, 2011.

Wang Q, Xu Y, Song JJ, Li CF, Ren PF, Xu JZ.: 3D stall delay effect modeling and aerodynamic analysis of swept-blade wind turbine.

660 Journal of Renewable and Sustainable Energy 5.6, 2013.

Whale J, Anderson CG, Bareiss R, Wagner S.: An experimental and numerical study of the vortex structure in the wake of a wind turbine. Journal of Wind Engineering and Industrial Aerodynamics 84.1, 1-21, 2000.

Wilson RE, Lissaman PBS.: Applied aerodynamics of wind power machines, Oregon State University, May 1974.

Wimshurst, A., Willden, R. H. J. : Computational observations of the tip loss mechanism experienced by horizontal axis rotors. Wind Energy,

665 21(7), 544-557, 2018.

Wimshurst, A., & Willden, R. H. J. : Analysis of a tip correction factor for horizontal axis turbines. Wind Energy, 20(9), 1515-1528, 2017.

Wood, D. H., Okulov, V. L., Bhattacharjee, D. : Direct calculation of wind turbine tip loss. Renewable Energy, 95, 269-276, 2016.

Wood, D. H., Okulov, V. L. :Nonlinear blade element-momentum analysis of Betz-Goldstein rotors. Renewable Energy, 107, 542-549, 2017.

670 Wood, D. H.: Including swirl in the actuator disk analysis of wind turbines. Wind Engineering, 31(5), 317-323, 2007.

Xiros MI, Xiros NI.: Remarks on wind turbine power absorption increase by including the axial force due to the radial pressure gradient in the general momentum theory. Wind Energy10: 99–102, 2007.

Zahle F, Sørensen NN.: On the influence of far-wake resolution on wind turbine flow simulations. Journal of Physics: Conference Series. Vol. 75. No. 1. IOP Publishing, 2007.